# Intron-mediated induction of phenotypic heterogeneity

Martin Lukačišin[1,2,4,5], Adriana Espinosa-Cantú[1,5] & Tobias Bollenbach[1,3 ✉]

Intragenic regions that are removed during maturation of the RNA transcript—introns—are universally present in the nuclear genomes of eukaryotes[1]. The budding yeast, an otherwise intron-poor species, preserves two sets of ribosomal protein genes that differ primarily in their introns[2,3]. Although studies have shed light on the role of ribosomal protein introns under stress and starvation[4–6], understanding the contribution of introns to ribosome regulation remains challenging. Here, by combining isogrowth profiling[7] with single-cell protein measurements[8], we show that introns can mediate inducible phenotypic heterogeneity that confers a clear fitness advantage. Osmotic stress leads to bimodal expression of the small ribosomal subunit protein Rps22B, which is mediated by an intron in the 5′ untranslated region of its transcript. The two resulting yeast subpopulations differ in their ability to cope with starvation. Low levels of Rps22B protein result in prolonged survival under sustained starvation, whereas high levels of Rps22B enable cells to grow faster after transient starvation. Furthermore, yeasts growing at high concentrations of sugar, similar to those in ripe grapes, exhibit bimodal expression of Rps22B when approaching the stationary phase. Differential intron-mediated regulation of ribosomal protein genes thus provides a way to diversify the population when starvation threatens in natural environments. Our findings reveal a role for introns in inducing phenotypic heterogeneity in changing environments, and suggest that duplicated ribosomal protein genes in yeast contribute to resolving the evolutionary conflict between precise expression control and environmental responsiveness[9].

Free-living cells are frequently challenged by changes in their environment, which results in a need to allocate their resources accordingly. One of the most important cellular processes to manage is the production of ribosomes[10,11]. Ribosome synthesis, although essential for growth and division, consumes a substantial fraction of cellular resources[12]. To assemble into functional complexes, the coordinated expression of tens of ribosomal RNA and ribosomal protein (RP) genes is necessary[13]. In the budding yeast, which underwent a whole-genome duplication followed by the loss or divergence of most duplicated genes[14], RPs are largely conserved as duplicates (ohnologues) with high sequence identity[2]. Why evolution may have favoured the retention of RP ohnologues remains an open question; possible reasons include increased gene dosage, genetic robustness towards mutations, and distinct biological roles or differential regulation of the duplicated genes[2,14–22]. Recently, it has been suggested that duplicated transcription factors, which are also preferentially retained, might be an evolutionary solution to a trade-off between tight regulation of expression level and responsiveness to environmental changes, such as stress[9]. In this scenario, one ohnologue provides a precise level of protein expression that is required irrespective of the environment, whereas the other one generates population heterogeneity, allowing for a flexible response when the environment changes. While the phenotypic effects of deleting the less-expressed copy of duplicated ribosomal genes are usually observed only under stress[17], it is unclear whether this paradigm applies to duplicated RPs.

The duplicated RP genes in yeast are highly enriched for introns. Although fewer than 5% of all yeast genes contain an intron[23], 94 out of 118 (80%) RP genes with an ohnologue do[3]. Differential expression of RPs through intronic regulation could expand their functional repertoire, even within the low sequence divergence; hence, introns may be involved in the evolutionary conservation of RP ohnologues[5,18]. Yeast introns in RP genes affect the expression level of the corresponding gene and, in some instances, that of its ohnologue[6,18,24–26]. Concerted intron retention in transcripts occurs in response to stress, suggesting that splicing regulation has a functional role in yeast[27]. Such a role is supported by the observation that intron deletion in the budding yeast results in growth alterations under conditions including drug treatment, starvation or population saturation[4–6]. Introns are thus clearly relevant for adapting to environmental changes; nevertheless, the determinants and functional outcomes of intron-mediated responses remain unclear.

As ribosome synthesis and protein translation are tightly coupled to growth rate[28], it is crucial to dissect cellular responses that are specific

[1]Institute for Biological Physics, University of Cologne, Cologne, Germany. [2]IST Austria, Klosterneuburg, Austria. [3]Center for Data and Simulation Science, University of Cologne, Cologne, Germany. [4]Present address: Faculty of Medicine, Technion – Israel Institute of Technology, Haifa, Israel. [5]These authors contributed equally: Martin Lukačišin, Adriana Espinosa-Cantú. ✉e-mail: t.bollenbach@uni-koeln.de

to particular stressors from responses to non-specific growth-inhibition when studying the effects of stress on ribosomal function. To this end, we previously established a method termed isogrowth profiling[7]. It is based on exposing cells to a combination of two drugs in an antiparallel concentration gradient discretized into separate liquid cultures, while keeping the overall inhibition constant. Analysing the transcriptome along the growth isobole then enables responses that are specific to each drug, or to their combination, to be distinguished from the general stress and growth inhibition responses.

Here, to investigate the role of introns and RPs in stress response, we extend isogrowth profiling from RNA sequencing (RNA-seq) to single-cell protein-level measurements. We found that lithium chloride (LiCl) induces extensive retention of introns in RP transcripts and an intron-dependent bimodal expression of Rps22B, a component of the small ribosomal subunit. The two subpopulations exhibit differential fitness under conditions of starvation and recovery. We show that whereas yeast in standard rich laboratory growth medium do not exhibit Rps22B bimodality, cells in medium with a high concentration of glucose do as they approach stationarity. Together, these results suggest that yeast has evolved an intron-mediated regulation mechanism of Rps22B to cope with uncertainty regarding the possible replenishment of nutrients at the end of exponential growth in a high-glucose environment.

## LiCl inhibits the splicing of RP transcripts

To study the effect of stress on the regulation of ribosomes, we started by investigating the transcriptional response to two growth inhibitors—LiCl, a pleiotropic drug that induces cationic and osmotic stress[29]; and cycloheximide, an inhibitor of the large ribosomal subunit[30]. As regulation of ribosomes is sensitive to changes in growth rate and medium composition[28], we used an antiparallel concentration gradient of the two drugs applied in separate liquid cultures[7], which consistently lowered the relative growth rate to near 50% (Fig. 1a, b). We inoculated the drug-free control at a lower cell density so that all samples reach a comparable cell density at the time of collection (Fig. 1b). To determine intron retention, RNA was extracted from the samples, ribo-depleted and sequenced (Methods). This procedure on the series of drug mixtures enabled us to observe the specific effects of stressors on intron retention in a way that is not confounded by growth-rate or cell-density effects.

We found that LiCl induces extensive intron retention compared to the drug-free control (Fig. 1c). Intron retention was not due to changes in growth rate or general stress, as no such increase was observed when cells were treated with cycloheximide (Extended Data Fig. 1a), or with two other drugs with different targets (Extended Data Fig. 2a). RP transcripts were the functional gene set most strongly affected by this increase in intron retention (Benjamini–Hochberg-corrected hypergeometric test, $P_{adj} = 3 \times 10^{-25}$, Fig. 1c, Extended Data Fig. 1b, c, Methods). The increase in intron retention was correlated with a decrease in their host transcript level (Extended Data Fig. 1d, e). Analysing RNA-seq reads that span the exon–intron junctions confirmed that introns are retained in mature transcripts, rather than being spliced out but not degraded (Extended Data Fig. 1f); this observation was further corroborated by observing the increase in intron retention also in the polyadenylated fraction of RNA (Extended Data Fig. 2b). It has previously been proposed that splicing of RP genes is downregulated by the accumulation of stable excised introns in the stationary phase[4]. However, these introns were not retained differently to other introns under LiCl stress (Extended Data Fig. 1g), suggesting that the high retention of introns in RP transcripts observed with LiCl treatment is not mediated by the stable linear excised introns previously reported.

## Osmotic stress induces Rps22B bimodality

*RPS22B* was a clear outlier with respect to intron retention under LiCl treatment. *RPS22B* encodes a component of the small ribosomal subunit

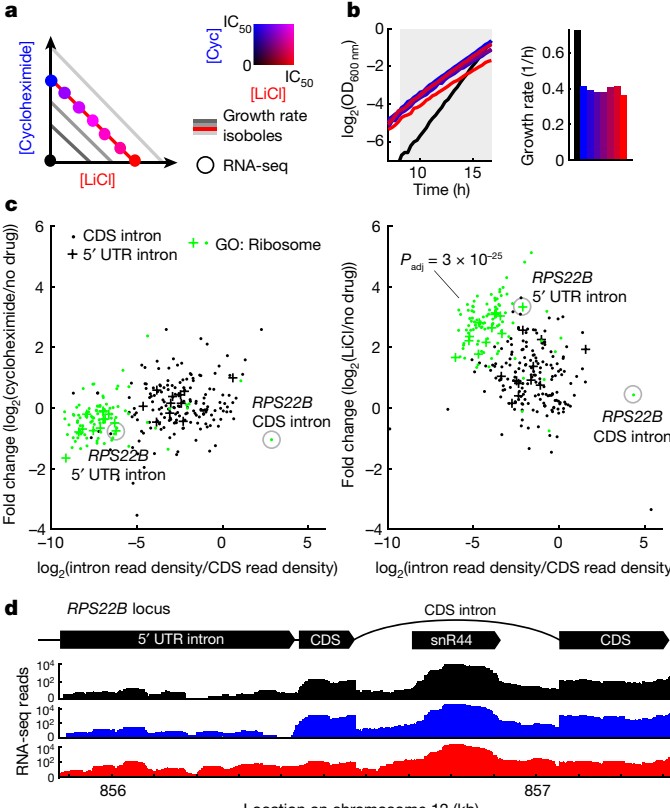

**Fig. 1 | LiCl-induced stress inhibits the mRNA splicing of ribosomal proteins. a**, Schematic of the isogrowth profiling that was used to characterize changes of splicing in mRNA. The coloured dots indicate the points in the two-drug gradient used to extract total RNA that was subsequently ribo-depleted and sequenced. $IC_{50}$, half-maximum inhibitory concentration. **b**, Growth curves of samples used for RNA-seq. The no-drug control was inoculated at a lower density to reach a comparable optical density at the time of extraction. The shaded area denotes measurements used to determine the exponential growth rates (right). Colour code as in **a**. $OD_{600 nm}$, optical density at 600 nm. **c**, Intron retention rate (intron read density/CDS read density; Methods) for all nuclear introns when treated with cycloheximide or LiCl, normalized to no-drug control (*y* axis) versus non-normalized (*x* axis). The most significantly enriched Gene Ontology (GO) cellular component gene set for the LiCl sample ('ribosome') is shown in green with the corresponding *P* value of a hypergeometric test. CDS introns are represented by dots and 5′ UTR introns by plus signs. See Extended Data Fig. 1 for data at other points of the isobole. **d**, RNA-seq counts for the *RPS22B* locus, colour-coded as in **a**. Top, schematic of the *RPS22B* introns and exons and *snR44* locus according to the *Saccharomyces* Genome Database.

and is one of only nine *Saccharomyces cerevisiae* genes containing two introns[31]. The intron in the 5′ untranslated region (5′ UTR) of *RPS22B* showed the highest increase in intron retention among 5′ UTR introns under osmotic stress (Fig. 1c), whereas there appeared to be numerous copies of the coding sequence (CDS) intron of *RPS22B*, presumably owing to a small RNA encoded within the intron that is under separate regulation (Fig. 1d; *snR44*). The prevalence of stop codons in the introns of *RPS22B* (Extended Data Fig. 3a, b), typical of yeast introns[32,33], suggested that intron retention in the transcript could manifest itself in altered protein levels rather than protein isoforms. To observe the single-cell behaviour of Rps22B at the protein level, we coupled the rigorously controlled combinatorial drug treatment to flow cytometry using a yeast strain with a GFP-tagged Rps22B[8]. Notably, Rps22B exhibited two clearly distinct levels of expression at intermediate LiCl concentrations (Fig. 2a). Other osmotic stressors similarly induced bimodal expression of Rps22B (Extended Data Fig. 3c).

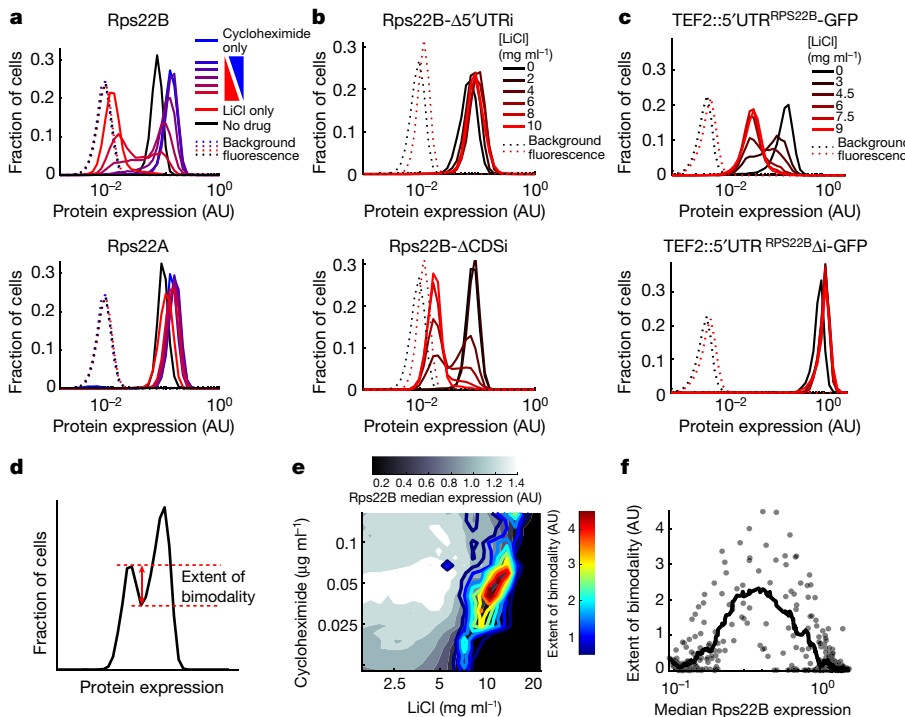

**Fig. 2 | The 5′ UTR intron mediates bimodal Rps22B protein expression under LiCl, consistent with a bistable regulatory loop. a**, Histograms of flow cytometry measurement of strains with GFP-tagged Rps22B or Rps22A. The RP gene *RPS22B*, which contains the 5′ UTR intron with the largest increase in retention due to LiCl, exhibits bimodal protein expression at intermediate LiCl concentrations, whereas its ohnologue, *RPS22A* that contains no introns, does not. See Extended Data Fig. 3 for Rps22B expression in other osmotic stresses. AU, arbitrary units. **b**, As in **a**, but with seamless deletion of either of the *RPS22B* introns (Δ5′UTRi, deletion of 5′ UTR intron; ΔCDSi, deletion of CDS intron). See Extended Data Fig. 3d for a strain with both introns deleted. **c**, As in **a**, but for strains with the 5′ UTR of *RPS22B* fused to *GFP*, with or without the 5′ UTR intron. **d**, Ad hoc definition of the extent of Rps22B bimodality (Methods). **e**, The extent of Rps22B bimodality (coloured lines) is overlaid on the median protein level of Rps22B (greyscale) in a two-drug gradient of LiCl and cycloheximide. **f**, Extent of Rps22B bimodality as a function of median Rps22B protein level is shown for all wells in the 2-drug gradient (dots) alongside the running average with a window of 30 data points (line). Rps22B bimodality peaks at a certain level of median Rps22B expression, rather than at a certain growth rate or LiCl concentration. See Extended Data Figs. 4, 5 for comparison with LiCl–myriocin drug pair.

## The 5′ UTR intron mediates Rps22B bimodality

Although LiCl induced bimodal expression of Rps22B, no such expression pattern was observed for Rps22A, the intronless ohnologue of Rps22B (Fig. 2a), the protein sequence of which is identical to that of Rps22B except for a single amino acid (Extended Data Fig. 3d). As the promoters of *RPS22A* and *RPS22B* behave similarly under different conditions[34], we hypothesized that the bimodal expression of Rps22B that is observed under osmotic stress depends on the presence of one or both introns; the 5′ UTR intron in particular is highly conserved both within Saccharomycetaceae[23] and in *S. cerevisiae* itself (Extended Data Fig. 3e). Deletion of the 5′ UTR intron, but not the CDS intron, abrogated the bimodal expression of Rps22B under LiCl stress (Fig. 2b, Extended Data Fig. 2f). In addition, it resulted in higher levels of *RPS22B* transcripts (Extended Data Fig. 3g) and protein (Fig. 2b, Extended Data Fig. 3f) under LiCl stress compared to the parental strain. No growth defects that could explain the differences in protein expression between the strains were observed (Extended Data Fig. 3h). This suggests that, in the absence of the 5′ UTR intron, cells under LiCl stress can no longer downregulate the production of Rps22B, consistent with previous reports that the double-stranded RNA structure contained in the intron is necessary for the regulated degradation of the transcript[35]. Fusing the 5′ UTR of *RPS22B* to *GFP* conferred bimodal GFP expression under LiCl stress, but only if the 5′ UTR intron was left intact (Fig. 2c). These results show that the 5′ UTR of *RPS22B* is a *cis*-regulatory element sufficient to confer bimodal protein expression to an unrelated gene, and corroborate that the evolutionarily conserved 5′ UTR intron is necessary for this effect.

To further elucidate the role of the 5′ UTR intron in the protein heterogeneity of Rps22B, we quantified intron retention in different situations. Fluorescence-activated cell sorting (FACS) of the LiCl-induced Rps22B-bimodal cells showed that Rps22B-high cells have lower intron retention and higher CDS transcript level; however, these differences alone appear insufficient to account for the pronounced bimodality in protein expression (Extended Data Fig. 3g). Moreover, sequencing the polyadenylated RNA fraction revealed that although LiCl induces a strong increase in 5′ UTR intron retention in the Rps22B transcript, this increase is not necessary for the induction of Rps22B protein bimodality by other osmotic stressors (Extended Data Fig. 3g). This suggests that the bimodal expression of Rps22B under osmotic stress is mediated by a post-transcriptional mechanism that requires the presence of the 5′ UTR intron and amplifies cell-to-cell differences in intron retention at the protein level.

## Rps22B expression is potentially bistable

A bimodal pattern of expression suggests—but does not necessarily imply—an underlying bistable regulatory circuit, such as a positive feedback loop[36]. A general hallmark of such a regulatory circuit is the existence of an unstable fixed point. Here, this fixed point would correspond to a protein level at which a small increase or decrease causes the cell to go to one of the two stable fixed points that correspond to high or low expression states[37], respectively. To address whether such an unstable fixed point exists for Rps22B, we examined the Rps22B bimodality in the presence of LiCl while using another drug to perturb the growth rate and, as measured, the overall level of Rps22B.

We used an automated liquid handling set-up that enables continuous monitoring of culture over a fine two-drug gradient distributed over six 96-well microtitre plates[7] (Extended Data Fig. 4a), and at the end of the incubation measured the single-cell protein expression by flow cytometry (Methods). We used LiCl in combination with two drugs with disparate mechanisms—the translation inhibitor cycloheximide and the sphingolipid synthesis inhibitor myriocin[38]. We found that the extent of how clearly the expression levels of the two subpopulations are separated into two peaks depends primarily on the median protein level of Rps22B (Fig. 2d, e, Extended Data Fig. 4c, d), with a maximum at an intermediate level (Fig. 2f, Extended Data Fig. 4e). This behaviour is consistent with the existence of an unstable fixed point at this protein level, causing what would be an otherwise log-normally distributed population centred around this point to divide equally into the two presumably stable subpopulations.

## Single-cell isogrowth profiling

To investigate the extent to which osmotic or translation stress induces distinct cellular states, as observed for Rps22B, we took advantage of the yeast protein–GFP library[8], which contains 4,156 strains with single protein–GFP fusions, and performed genome-wide single-cell isogrowth profiling (Extended Data Fig. 6a–c). We profiled the entire library in four of the conditions used previously (Supplementary Table 1) and selected strains that, on visual inspection, did not show a clearly unimodal expression pattern for further characterization using a more detailed antiparallel gradient (Supplementary Table 2). We found several instances of protein expression heterogeneity resulting from non-specific growth inhibition (Extended Data Fig. 6d) or specifically from either of the stresses (Extended Data Fig. 6e). Cycloheximide induced bimodal expression of Hsp12, the budding yeast persistence marker[39], and of Aro9, an enzyme involved in production of the yeast quorum-sensing molecule tryptophol[40], a trigger for invasive growth in low-nitrogen environments[40]. LiCl induced heterogeneity in Rps9A, another small RP subunit that contains an intron in its gene. Similar to Rps22B, the stress-induced heterogeneity of Rps9A levels was intron-mediated (Extended Data Fig. 6f). In addition, there was a general trend in which an increase in intron retention in LiCl correlated with a decrease in the level of the respective protein (Extended Data Fig. 6g). Overall, these observations indicate that intron retention is used by the yeast cell to control the protein level and, in the case of Rps9A and Rps22B, also the protein level heterogeneity.

## Differential fitness during starvation

Introns have a key role in preparing the yeast population for starvation[5,27]. Therefore, we hypothesized that the two subpopulations, defined by distinct levels of Rps22B expression, have differential fitness under starvation stress. To test this idea, we subjected an exponentially growing culture of the Rps22B–GFP strain to LiCl stress using a microfluidic system (Methods). After the establishment of bimodality, we replaced the LiCl-containing growth medium with spent medium. In this way, we induced starvation of varying duration, followed by a sudden switch to rich medium (Fig. 3a). At first, the spent medium triggered a rapid disappearance of the strong nucleolar signal of Rps22B–GFP (Supplementary Videos 1, 2), which was presumably caused by the redistribution of nucleolar proteins across the cytoplasm upon stress, as occurs in other model systems[41]. After the switch to rich medium, the nucleolar signal of Rps22B re-emerged in cells that started growing again. Notably, the fitness of cells depended on their levels of Rps22B expression immediately before the starvation stress. Longer starvation stress favoured the survival of cells that expressed Rps22B at low levels, in that they lysed less frequently than the highly expressing cells. By contrast, shorter starvation favoured the cells that expressed Rps22B at high levels, in that they budded more in the period following

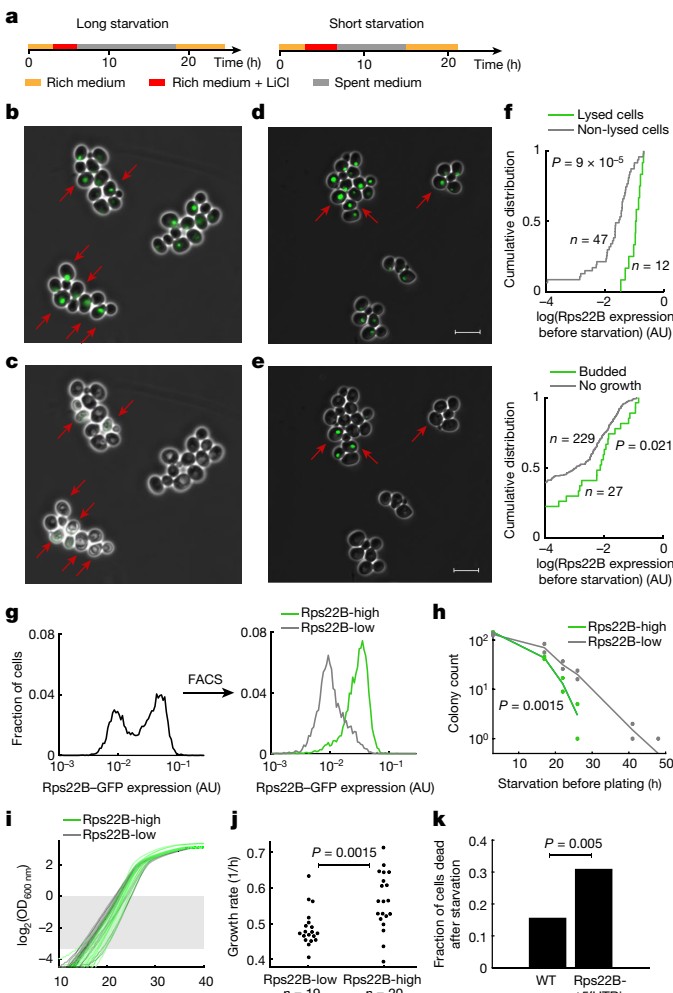

**Fig. 3 | Rps22B expression level confers a selective advantage under starvation stress. a**, Schematic of the temporal sequence of media used in microfluidic microscopy experiments. **b**, **c**, Example micrographs of Rps22B–GFP yeast cells after the application of LiCl, showing LiCl-induced bimodality of Rps22B expression (**b**). Rps22B-high cells (dark red arrows) lyse after prolonged starvation and nutrient replenishment (**c**). **d**, **e**, As in **b**, **c**, but for a shorter period of starvation. After a shorter period of starvation and medium replenishment, the Rps22B-high cells marked with dark red arrows (**d**) budded more than other cells (**e**) (Supplementary Videos 1, 2). Scale bars, 10 μm. **f**, Quantification of the time-lapse micrographs. Top, cumulative distribution of Rps22B expression measured before starvation in cells that have either lysed (green) or not (grey) within 6 h after medium replenishment following a long starvation. Bottom, cumulative distribution of Rps22B expression before starvation in cells that have either budded (green) or not (grey) within 6 h after medium replenishment following a shorter starvation. Rps22B expression values between the two panels are not comparable. A two-sided Mann–Whitney $U$ test was used to determine significance. **g**, Histograms of Rps22B–GFP expression in LiCl before and after FACS. **h**, Survival curves (colony-forming units) of the sorted populations as a function of time spent in phosphate-buffered saline (PBS) after the sorting and before plating on rich medium. The two-sided bootstrapped $P$ value is shown (Methods). **i**, Growth curves of cultures in rich medium after sorting. The shaded rectangle denotes the optical density range that was used to quantify growth rates. **j**, Quantification of growth rates from **i**. Significance was determined using a two-sided $t$-test. **k**, Fraction of cells that were dead after starvation in time-lapse microscopy experiments comparing wild-type (WT) Rps22B and the *RPS22B* 5′ UTR intron deletion mutant. For experimental set-up and further quantification, see Extended Data Fig. 7. Significance was determined using a one-sided permutation test.

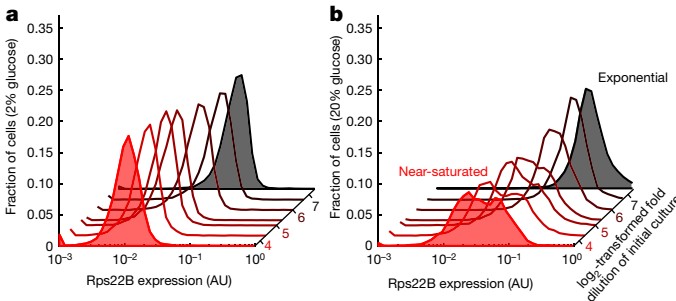

**Fig. 4 | A high-glucose environment induces bimodal Rps22B expression as cells near saturation. a**, **b**, Histograms of Rps22–GFP expression as measured by flow cytometry after 10 h of incubation in standard rich medium (2% (w/v) glucose; **a**), or in a yeast peptone medium supplemented with 20% (w/v) glucose (**b**). The red hue indicates increasing cell density in the initial inoculum. Examples of exponential and near-saturated cultures are highlighted by shading.

nutrient replenishment (Fig. 3b–f). FACS corroborated this phenotypic difference, as the Rps22B–GFP-high subpopulation exhibited poorer survival under starvation (Fig. 3h), but a higher growth rate in rich medium (Fig. 3i, j) compared to the Rps22B–GFP-low subpopulation. This marked diversification of the population observed for Rps22B, with clear fitness effects under a subsequent stress, may reflect a bet-hedging strategy[42–44].

To test whether the link between intron-mediated Rps22B protein heterogeneity and phenotypic heterogeneity is causal, we performed time-lapse microscopy with the *RPS22B* 5′ UTR intron deletion mutant (Extended Data Fig. 7a, Supplementary Videos 3, 4). Deletion of the 5′ UTR intron in *RPS22B* not only increased the fraction of Rps22B–GFP-high cells, but also the fraction of cells that died under starvation (Fig. 3k, Extended Data Fig. 7b, d). Notably, it also partially abolished the coupling between phenotypic heterogeneity and Rps22B protein heterogeneity (Extended Data Fig. 7e), suggesting that downregulation of Rps22B specifically through the intron-mediated mechanism is important for phenotypic benefit under starvation. Population-level assays with the intron-deletion mutant confirmed that its survival times under starvation are less heterogeneous and lower on average (Extended Data Fig. 8a), and that its growth rate is increased under LiCl stress (Extended Data Fig. 8b). These observations confirm that the *RPS22B* 5′ UTR intron can mediate not only Rps22B expression heterogeneity, but also phenotypic heterogeneity.

## Bimodal Rps22B expression in high-glucose conditions

Bet-hedging strategies can evolve if the eliciting signal is probabilistically followed by stress in which the subpopulations exhibit differential fitness in the natural environment[42,45]. However, yeasts nearing saturation, and hence starvation, in standard rich laboratory medium do not exhibit Rps22B bimodality (Fig. 4a). Therefore, we wondered whether there is a plausible natural setting in which Rps22B bimodality is triggered just before the onset of starvation. We reasoned that yeasts may often be exposed to hyperosmotic sugar concentrations, such as those in ripe fruits. High glucose elicits osmotic responses that are different from those to salt[46]. In addition, a high concentration of glucose, despite being an osmotic stressor, leads to comparatively higher expression levels of RPs than LiCl stress (Extended Data Fig. 9). As Rps22B exhibits the maximum extent of bimodality at a certain median expression level (Fig. 2d–f), we hypothesized that a high-glucose environment might specifically induce Rps22B bimodality as yeast nears saturation, when RP levels decrease owing to growth slowdown.

To test this idea, we inoculated Rps22B–GFP cultures into a yeast peptone medium with a high concentration of glucose (20% w/v)

that emulates the total hexose concentration in ripe grapes[47], at various initial cell densities. Exponentially growing cells in high-glucose medium generally resulted in less clear bimodal expression of Rps22B–GFP compared to cells treated with LiCl; however, cultures in high-glucose medium that were nearing saturation exhibited a pattern of Rps22B expression that was clearly bimodal (Fig. 4b). This observation suggests that part of the yeast population is attuned to a probabilistic event of nutrient replenishment that can follow growth in a high-glucose environment, while the rest of the population is preparing for starvation.

## Discussion

In the budding yeast, the overall expression level of RPs results from two sets of RP genes with high amino-acid sequence similarity, which often differ in the presence and identity of intronic sequences[6,18]. Here, we have uncovered an intron-mediated regulation of protein expression heterogeneity. We showed that LiCl leads to a widespread retention of introns[48] that is independent of growth rate perturbations and predominantly affects RP transcripts. The small ribosomal subunit protein gene *RPS22B* manifested bimodal protein expression under conditions of osmotic stress, and this effect was mediated by its 5′ UTR intron, which is conserved throughout the Saccharomycetaceae[23]. By contrast, its intronless ohnologue *RPS22A* exhibited a unimodal pattern of protein expression irrespective of stress. This behaviour of the *RPS22* gene pair offers a paradigm to explain the function of introns in duplicated RP genes, in that they enable differential and versatile regulation for duplicated genes as an evolutionary trade-off between environmental responsiveness and precise regulation[9].

We observed population diversification with respect to starvation stress and subsequent recovery, which was marked by Rps22B expression levels and partially dependent on the *RPS22B* 5′ UTR intron. Reminiscent of bet-hedging strategies[42], we found that in high-glucose medium, yeast populations diversify in their expression of Rps22B as they enter the stationary phase, as if anticipating a probabilistic replenishment of nutrients. Why such a behaviour should have evolved for a high-glucose environment, but does not manifest at glucose levels present in the standard laboratory 'rich' medium, is unclear. One plausible scenario is that for yeast living on the skin of grapes, a high concentration of glucose is an environmental signal that the fruits are ripening and bursting. Osmotic bursting of fruits is known to follow a probabilistic trajectory over time[49] and is known to happen predominantly during rainfall[50]; hence, in a cluster of grapes, vigorous yeast growth due to bursting of one of the berries might be followed by nutrient replenishment due to bursting of a neighbouring berry, determined by the probabilistic aspects of rain duration and intensity.

The fact that we found osmotic-stress-induced, intron-mediated phenotypic heterogeneity only for Rps22B and Rps9A, although most other pairs of duplicated RP genes also contain at least one intron, is notable, especially as the effect of introns on protein expression level has been reported for other RPs in the budding yeast[6]. It is thus tempting to speculate that for other RPs, there might exist other levels of osmotic stress—or other stressors altogether—that would trigger population diversification with respect to the expression level of a given RP. Such a set of diversification mechanisms could then present a versatile stress toolkit for the yeast population, enabling yeast to brace against continued stress and at the same time maintaining a small, stress-sensitive subpopulation that is poised to rapidly exploit a short window of fitness advantage should the stress suddenly disappear. Our study thus highlights the need for further study of the intronic regulation of population diversification, and demonstrates the utility of using graded, growth-rate controlled perturbations.

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

# Methods

## Transcriptional isogrowth profiling

For isogrowth RNA-seq, *S. cerevisiae* strain BY4741 was grown in seven conditions with varying ratios of LiCl (Sigma-Aldrich, L9650) and cycloheximide (Sigma-Aldrich, 37094), ensuring 50% growth inhibition, and in YPD (yeast extract (Sigma-Aldrich Y1625) 1% w/v, peptone (Sigma-Aldrich 91249) 2% w/v, dextrose (Sigma-Aldrich D9434) 2% w/v) containing no drug (Extended Data Table 1). The frozen stock was diluted 130 times into drug-containing wells and 100 times more into wells without drug. The cells were incubated for a total of around 17 h to a final absorbance of around 0.1 on the 96-well plate, corresponding to an $OD_{600 nm}$ of approximately 0.5. Cells were then collected and the RNA extraction was performed using the RiboPure RNA Purification Kit for yeast (Thermo Fisher Scientific, AM1926). The extracted RNA was sent to the Next Generation Sequencing Facility of the Vienna Biocenter Core Facilities, where it was rRNA depleted using the Illumina Ribozero Yeast Kit, multiplexed using Illumina True Seq adapters, single-end 50-bp sequenced using Illumina HiSeqV4 and demultiplexed.

RNA samples from treatments with single stressors were obtained from cultures with automated re-inoculation every 8 h (ref. [7]), with a total incubation time of 24 h, to keep cultures in exponential phase and ensure the drug effects have taken place. Strains were grown in YPD with a drug, either in a concentration gradient to select samples closest to 50% of growth inhibition (LiCl, fenpropimorph and myriocin), or in a single concentration (0.4 M NaCl and glucose 20%). Fenpropimorph (Sigma-Aldrich 36772) was tested in a 0.5–1.5 μM gradient; sequenced samples were treated with 0.85 μM fenpropimorph. The myriocin (Sigma-Aldrich 476300) gradient encompassed 0.25–0.75 μg ml$^{-1}$; 0.32 μg ml$^{-1}$ myriocin samples were sequenced. For each sample of extracted RNA, both mRNA and ribo-depleted total RNA sequencing libraries were prepared at the Cologne Center for Genomics (CCG) and sequenced using 2 × 100-bp paired-end reads.

## Intron retention analysis

The reads resulting from sequencing were aligned to the annotated reference *S. cerevisiae* genome R64-2 using TopHat[51] or HISAT2[52]. The reads mapping to introns or CDS of intron-containing genes were quantified using featureCounts[53] using custom produced .saf files extracted from the reference .gff annotation file. The intron retention rate was calculated as the ratio of intron counts normalized to the intron length over the CDS counts normalized to the length of the CDS region. For genes containing multiple introns, the introns were treated jointly for CDS introns and separately for 5′ UTR introns. Alignments were visualized using Integrative Genomics Viewer[54].

For the analysis shown in Extended Data Fig. 1f, first the number of contiguous reads (no mismatches allowed) overlapping the individual intron ends was determined. This count was then averaged between the 5′ and the 3′ end of the intron ends and divided by the read length (50 bp). The intron retention rate was then calculated as the ratio of the resulting value over the CDS counts normalized to the length of the CDS region.

For GO enrichment analysis, intron-containing genes were ordered into a ranked list on the basis of the fold change increase in intron retention in LiCl compared to the no-drug control, either in decreasing (Fig. 1c) or increasing (Extended Data Fig. 1b) order. The ranked list was then analysed using GOrilla[55], for either cellular component or biological process, respectively. The adjusted *P* values reported here correspond to the false discovery rate reported by GOrilla. Gene enrichments were visualized using REVIGO[56].

## Rps22B intron conservation analysis

To gauge the evolutionary conservation of the Rps22B 5′ UTR intron within the natural population of *S. cerevisiae* (Extended Data Fig. 3e), previously published sequencing data for 1,011 isolates[57] were multiple-sequence aligned using Clustal Omega[58] and visualized using MSA-BIOJS[59].

## Screening the entire yeast protein–GFP library for drug-induced bimodality

The whole yeast protein–GFP library[8], contained in 43 96-well microplates, was screened for drug-induced bimodality by culturing in four conditions: YPD medium containing no drug, cycloheximide, LiCl, or a combination of LiCl and cycloheximide. The screen was divided into six experimental batches. In every batch, up to nine plates of the library were processed, with the same strain being cultured in four conditions on the same day and on the same microtitre plate, giving rise to up to 36 microtitre plates per experiment. Before the experiment, empty microtitre plates and YPD medium were pre-incubated at 30 °C to ensure reproducibility of gene expression measurements. LiCl, cycloheximide or a mixture of the two were diluted with YPD to a final concentration needed to achieve 40% inhibition and ensuring that in the mixed condition, the LiCl and cycloheximide were mixed in an equipotent manner, meaning that on their own they both elicited approximately the same (smaller) growth inhibition (Extended Data Table 1). The drug solutions were pipetted row-wise into a 96-well microtitre plate, including a drug-free YPD control, enabling the cultivation of 24 strains in 4 conditions per microtitre plate. The library plates stored at −80 °C were thawed and, using an electronic multi-step multi-channel pipette, 1.5 μl saturated glycerol stock of the corresponding strain was inoculated into each drug-containing well and 0.2 μl of the saturated glycerol stock was inoculated into drug-free YPD as a control. The smaller inoculum size for the drug-free control was designed to ensure that the final cell density at the end of the experiment was comparable to that of the drug-containing wells. Plates were incubated for around 14 h in an automated incubator (Liconic StoreX) kept at 30 °C and greater than 95% humidity, vigorously shaken at more than 1,000 rpm. During the incubation, $OD_{600 nm}$ was measured every approximately 45 min in a Tecan Infinite F500 plate reader. In addition to shaking during incubation, directly before each measurement, plates were shaken on a magnetic shaker (Teleshake; Thermo Fisher Scientific) at 1,100 rpm for 20 s. The automated set-up was programmed using the Tecan FreedomEvo software. The growth rates were inferred by fitting a line to the log-linear part of $OD_{600 nm}$ measurements between 0.01 and 1.

After incubation, the yeast cells in 96-well microtitre plates were twice centrifuged at 1,050*g* for 3.5 min and resuspended in ice-cold Tris-EDTA buffer by vigorous shaking at 1,000 rpm on a Titramax shaker for 30 s. After another centrifugation at 1,050*g* for 3.5 min, the cells were resuspended in 80 μl of Tris-EDTA and immediately stored at −80 °C. On the day of the flow cytometry measurement, the plates were thawed on ice for around 3 h and kept on ice until the measurement. The fluorescence was measured for 10,000 cells using BD FACS Canto II equipped with a high-throughput sampler, using FACS Diva software. The green fluorescence measured in FITC-H channel was normalized to the forward scatter FSC-H channel.

Strains exhibiting bimodality, minor peaks or prominent changes in gene expression at visual inspection of the expression histograms were manually selected for a more detailed screen. The selected strains were re-streaked from the yeast protein–GFP library microtitre plates onto YPD-agar plates and single clones for each strain were picked and cultured in a 96-well microtitre plate to saturation. Glycerol was added to a final concentration of 15% and the plates were frozen. The plates were screened in eight conditions (Extended Data Table 1) and analysed using flow cytometry in a way similar to that described above. The identity of the relevant strains—Rps22A, Rps22B, Aro9, Hsp12, Cit1 and Tdh1—was confirmed by PCR and gel electrophoresis as reported before[8], using a common F2CHK reverse primer and strain specific oligos taken from (https://yeastgfp.yeastgenome.org/yeastGFPOligoSequence.txt). The strain Smi1 appeared bimodal in the screen; however, its identity could not be confirmed as above. The identity of other strains was not tested.

## Construction of GFP-labelled intron deletion strains and flow cytometry experiments

The seamless intron deletion strains for Rps22B (Rps22B-Δi1-Δi2, rps-22bDi_JPY138F4; Rps22B-Δi1, rps22bD1i_MDY125A4; Rps22B-Δi2, rps-22bD2i_MDY125A9) and Rps9A (Rps9A-Δi, rps9aDi_MDY133H8), as well as the parental haploid strain WT_JPY10H3 (MATa ura3Δ0 lys2Δ0 leu2Δ0 his3Δ200) were a gift from J. Parenteau and S. Abou Elela[5]. The parental and intron deletion strains were labelled with GFP fused to the protein of interest following homologous recombination of the PCR-amplified fluorescent marker from the matching yeast protein–GFP library strains. To achieve this, the DNA of the Rps22B–GFP and Rps9A–GFP strains from the library was extracted using the Yeast DNA Extraction Kit (Thermo Fisher Scientific, 78870). PCR amplification of the GFP label with 40 base pairs flanking sequences was done using the corresponding F2 and R1 pair of primers from a previous study[8] (https://yeastgfp.yeastgenome.org/yeastGFPOligoSequence.txt) in the following reaction conditions: 2 ng ml[−1] DNA in a final mix of 20 μl, with 0.4 units of Phusion High-Fidelity DNA polymerase (Thermo Fisher Scientific, F530SPM), 4 μl of 5× Phusion HF buffer, 0.2 μM dNTPs and 0.5 μM primers. PCRs were incubated for 3 min at 98 °C, followed by 34 cycles of 98 °C for 30 s and 72 °C for 2.5 min, and a final incubation at 72 °C for 5 min. The resulting PCR products were transformed into the parental WT_JPY10H3 and Δi strains as in a previous report[60], selecting on SD-HIS agar plates (Takara Bio, 630411; Diagonal GmbH&CoKG, Y1751). The correct genomic context insertion of GFP of single colonies was confirmed by PCR using a combination of primers that also allowed the confirmation of the intron deletion strain. To confirm the Rps22B–GFP fusions in the Rps22B-Δi and WT_JPY10H3 strains, we used primers rps22b-899F 5′-CCGTTATTCTTCTCGCAACC-3′ binding upstream of the 5′ UTR intron, and RPS22B-CHK (ref. [60]) 5′- ACTAGATGGTGTGATCGGGC-3′ binding in the CDS intron sequence in combination with the reverse primer F2CHK (ref. [60]) 5′- AACCCGGGGATCCGTCGACC-3′, complementary to the GFP sequence. To confirm the Rps9A–GFP fusions in the Rps9A-Δi and WT_JPY10H3 strains, we used rps9a-800F 5′-GTTCGATTTCTTGGTCGGACGC-3′ upstream of the open reading frame and F2CHK. Once the successful construction of the GFP reporter strains was confirmed, 20 μl of saturated overnight cultures were inoculated into 96-well microtitre plates containing 180 μl of YPD with the respective concentration of LiCl. The microtitre plates were incubated at 30 °C and continuous shaking at 900 rpm on a Titramax shaker for 7 h, collected and measured on a flow cytometer as described above. Strains with genome-integrated GFP with different 5′ UTR fusions were a gift from the laboratory of G. Stormo[26].

## Measurement of Rps22B expression in detailed two-dimensional drug gradients

The re-inoculation set-up, as reported previously[7], in conjunction with flow cytometry measurements was used to measure Rps22B expression in detailed two-dimensional (2D) gradients of LiCl and cycloheximide, and LiCl and myriocin (Sigma-Aldrich, M1177). In brief, a *S. cerevisiae* strain with the *RPS22B* gene fused to GFP protein from the ORF-GFP library[8] was grown in YPD broth in a conical flask overnight and then distributed into a 96-well plate. A customized robotic setup (Tecan Freedom Evo 150) with eight liquid handling channels and a robotic manipulator was used to produce a 2D discretized two-drug 24 × 24-well gradient in YPD spread over 6 96-well plates and to inoculate the yeast overnight culture to a final liquid volume in the well of 200 μl and a final absorbance 0.03 on the 96-well plate, corresponding to a standardized OD$_{600\,nm}$ of 0.15. Working drug solutions were prepared either by adding the respective amounts of concentrated DMSO drug stocks thawed from −20 °C storage (no refreezing) previously prepared from stock chemicals (cycloheximide and myriocin), or by dissolving directly in YPD and sterile-filtering (LiCl). The six plates were incubated for three iterations, each lasting around 8 h. After the incubation, the cells were collected and measured using flow cytometry as described above. Rps22B protein bimodality was quantified as the depth of the trough on the Rps22B protein single-cell expression histogram; that is, the *y*-axis distance between the trough and the lower of the two peaks (Fig. 2d). To this end, the MATLAB function findpeaks was used to determine the prominence of the peak that was created from the trough through inversion of the histogram values.

## FACS of Rps22B–GFP and experiments with GFP-sorted cells

Constitutive cytosolic mCherry expression was added to the parental and intron deletion GFP-fusion strains. To do so, a *TDH3*::mCherry construct was inserted in the HO locus by homologous recombination after transformation of a NotI-digested SLVA06 plasmid[61], provided by the laboratory of M. Springer. The Rps22B–GFP strain with constitutive mCherry expression was inoculated into YPD and incubated at 30 °C for 16 h. The overnight culture was diluted 20-fold into 4.5 mg ml[−1] LiCl in YPD, incubated for 6 h and washed twice with PBS. Positive mCherry cells were sorted into low-GFP and high-GFP populations with a Becton Dickinson INFLUX cell sorter (for gating strategy, see Supplementary Fig. 1). Around $8 \times 10^6$ sorted cells of each condition were used for RNA extraction and sequencing (see 'Transcriptional isogrowth profiling'). Samples of sorted cells at a final concentration of 4,000 cells per ml were prepared with PBS for the following experiments. Immediately after sorting, 15 μl of the samples were inoculated into fresh YPD medium in 24 replicates each. Optical density was measured every 20 min for 48 h in a plate reader incubated at 30 °C with constant shaking. OD was corrected by subtracting the background, and growth rates were inferred by a linear fit to the log-linear part of the corrected OD measurements between 0.02 and 0.2. Growth rates higher than 0.85 h[−1], which resulted from bacterial contaminations, were discarded. A two-tailed *t*-test was performed to compare the growth-rates from the two sorted populations (using the function ttest2 in MATLAB R2019a). For scoring survival after starvation, sorted cells were incubated in PBS at 30 °C with shaking at 200 rpm. At each sampling time point, 50 μl of starving cultures were spread onto an OmniTray plate (Thermo Fisher Scientific) filled with 35 ml of YPD-agar, in two replicates for each sorted population. Agar plates were cultivated at 30 °C, pictures were taken after around 36 h, and colonies were counted using ImageJ. To quantify the difference between low-GFP and high-GFP populations (Fig. 3h), we used the difference Δ between the total low-GFP and high-GFP colony counts along all timepoints. To test the statistical significance of Δ, we used bootstrapping. In brief, we created 10[4] surrogates (artificial datasets) complying with the null hypothesis that there is no difference between the populations. To create one such surrogate, we sampled— for each replicate and time point—the colony count from the Poisson distribution with a rate λ corresponding to the mean colony count for that time point. We determined the *P* value as the fraction of surrogates that exhibit a higher Δ than the original experimental dataset.

## Time-lapse imaging of the Rps22B–GFP strain during starvation

For micrographs shown in Fig. 3, the ORF-GFP library Rps22B–GFP strain was used; for experiments with the intron deletion mutants (Extended Data Fig. 7), we used the 5′ UTR intron-deletion strain and its parental Rps22B–GFP strain, both transformed with a plasmid constitutively expressing mCherry (see 'FACS of Rps22B–GFP and experiments with GFP-sorted cells') to allow for easy tracking of cellular integrity, because cells lose the mCherry signal after lysis; this approach has been used previously for tracking the lysis of bacterial cells[62]. Strains were inoculated into YPD at 30 °C with shaking overnight. The resulting culture was diluted 20-fold and loaded to the CellASIC ONIX2 haploid yeast microfluidic plates (MerckMillipore) at 55 kPa pressure, using the CellASIC ONIX2 system software. A sequence of YPD, LiCl-YPD, starvation medium (spent medium or PBS) and YPD were flushed through the microfluidic chamber in a time-controlled manner at 10-kPa pressure. Spent medium was produced by incubation of the Rps22B–GFP strain

in YPD medium for seven days with subsequent sterile filtration. Starvation times were empirically selected such that an intermediate number of cells died in the long starvation condition or budded after the short starvation. The specimen was imaged every 5 min at multiple locations with a Nikon Eclipse Ti inverted microscope through a 100× oil objective in a 30-°C cage incubator, using NIS Elements software. The micrographs of yeast from just before the starvation stress (Fig. 3) were segmented and fluorescence-quantified using the CellStar MATLAB plug-in[63] and the fluorescence readout was $\log_{10}$-transformed. The time-lapse images were then visually scored to determine which cells budded or lysed after the replenishment of nutrients. The time-lapse videos from the experiments with the intron-deletion mutant were segmented and tracked using YeaZ[64].

### RPS22B 5′ UTR intron-deletion mutant fitness experiments

The 5′ UTR intron-deletion strain and its parental strain, with Rps22B–GFP fusion and constitutive mCherry expression, were grown overnight in YPD. For growth rate determination experiments in liquid media, 5 ml of each saturated overnight culture was standardized to $OD_{600\,nm} = 0.1$ and inoculated in a 1/10 dilution into YPD containing 0, 2.25, 4.5, or 9 mg ml$^{-1}$ LiCl in 12 replicates of each strain and condition. OD was measured every 5 min for 24 h in a plate reader during incubation at 30 °C with constant shaking. OD and growth rates were processed as above (see 'FACS of Rps22B–GFP and experiments with GFP-sorted cells'). For scoring survival under starvation, 1 ml of each overnight culture was inoculated into 19 ml of YPD with 9 mg ml$^{-1}$ LiCl. Cultures were incubated at 30 °C with 200 rpm shaking for 6 h. Cultures were standardized to $OD_{600\,nm} = 1$ and washed twice by centrifuging at 1,050 g for 5 min and resuspending pellets in 20 ml of PBS. Dilutions of 5,000 cells per ml were incubated at 30 °C with constant shaking. Plating onto YPD-agar, incubation and analysing colony counts were done as explained above (see 'FACS of Rps22B–GFP and experiments with GFP-sorted cells').

### Rps22 bimodality in other osmotic stresses and in high glucose

The ORF-GFP library Rps22B–GFP cultures were inoculated into YPD medium, YPD containing 0.6 M NaCl, 2 M KCl (Sigma Aldrich, S3014 and P9541, respectively) or into a yeast peptone medium with 5%, 10% or 20% (w/v) glucose (Sigma Aldrich, G8270). To measure the fluorescence intensity of cells from different cell densities and growth stages, six microtitre plates were prepared, each with the same conditions but with a gradient of initial inoculum sizes; in this way, each plate accounted for the stress array with different cell density and could be incubated and analysed at different time points. To achieve this, saturated overnight cultures were diluted with YPD in a 4/5 serial dilution (that is, 4 ml was transferred to the final volume of 5 ml) to obtain $(4/5)^0$ to $(4/5)^{23}$ dilutions. Subsequently, 15 µl of cell dilutions were inoculated in 185 µl of medium. Plates were incubated in the Liconic StoreX and kept at 30 °C and greater than 95% humidity, with constant shaking. Plates were processed at 8, 12, 16, 20 and 24 h after inoculation with an automated robotic system consisting of a ACell System (HighRes Biosolutions) integrated with a Lynx liquid handling system (Dynamic Devices), a plate reader (Synergy H1, BioTek) and a CytoFLEX flow cytometer (Beckman Coulter). First, plates were shaken and measured as described above to obtain the $OD_{600\,nm}$ values. Cultures were then transferred to a new plate and, when necessary, diluted to an absorbance of around 0.3 with Tris-EDTA buffer to avoid high cell densities, and measured by flow cytometry.

### Reproducibility

Rps22B protein bimodality in LiCl was replicated independently by two researchers, at least five times, using two different yeast strains; all attempts at replication were successful. Rps22B protein bimodality in NaCl and KCl was determined twice; all attempts at replication were successful. The loss of bimodality in the *RPS22B* 5′ UTR intron

deletion strain was replicated three times; all replication attempts were successful. The bimodal expression of the *RPS22B* 5′ UTR intron–GFP fusion in LiCl was observed twice; all attempts were successful. The Rps22B bimodality on the entry to the stationary phase after growth in high glucose was reproduced twice; the exact timing of observing maximum bimodality varied, so multiple time points were measured as described in 'Rps22 bimodality in other osmotic stresses and in high glucose'. All attempts at replication using the incubation method described were successful; one attempt at replication using a different microtitre plate shaker (Titramax 1000, Heidolph) resulted in less pronounced heterogeneity, which is likely to be due to a different degree of aeration or shaking. Intron retention determined by sequencing was confirmed independently by two researchers, once by each; all attempts at replication were successful. Time-lapse microscopy phenotype of Rps22B-high and Rps22B-low cells was replicated twice in slightly modified set-ups and with independent strains as detailed in Fig. 3, Extended Data Fig. 7; all attempts at replication were successful. For the experiment shown in Extended Data Fig. 7, two starvation conditions, in which the length of the starvation period was varied, were tested to achieve an intermediate number of cell deaths within the time frame of the experiment as assessed visually and the chosen condition was then quantified. The whole yeast–GFP library isogrowth scan was performed once owing to the scale of the experimental effort; about 10% of strains that did not show clear unimodal expression on visual inspection of the expression histograms were restreaked to exclude the possibility of contaminations and the measurement was repeated once on a more detailed antiparallel gradient as detailed in the manuscript; the bimodality of Rps9A and Aro9 was additionally replicated once in a detailed discretized 2D drug concentration gradient similar to the one shown in the manuscript for Rps22B. To determine the phenotypic effect of the Rps22B expression level, cells were sorted once and survival and growth assays performed in 2 and 24 replicates, respectively; the phenotypic effect was independently confirmed by other methods (time-lapse microscopy and phenotypic assays using the intron-deletion mutant in Extended Data Fig. 8).

### Reporting summary

Further information on research design is available in the Nature Research Reporting Summary linked to this paper.

### Data availability

The RNA-seq data are available in the Gene Expression Omnibus under accession numbers GSE155060 and GSE197174. The processed flow cytometry data from isogrowth profiling are available as Supplementary Tables 1 and 2 accompanying this manuscript. The *S. cerevisiae* R64-2 reference genome and annotation was downloaded from the National Center for Biotechnology Information, assembly reference GCF_000146045.2.

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

**Acknowledgements** We thank the IST Austria Life Science Facility, the Miba Machine Shop and M. Lukačišinová for support with the liquid handling robot; the Bioimaging Facility at IST Austria, J. Power and B. Meier at the University of Cologne, and C. Göttlinger at the FACS Analysis Facility at the Institute for Genetics, University of Cologne, for support with flow cytometry experiments; L. Horst for the development of the automated experimental methods in Cologne; J. Parenteau, S. Abou Elela, G. Stormo, M. Springer and M. Schuldiner for providing us with yeast strains; B. Fernando, T. Fink, G. Ansmann and G. Chevreau for technical support; H. Köver, G. Tkačik, N. Barton, A. Angermayr and B. Kavčič for support during laboratory relocation; D. Siekhaus, M. Springer and all the members of the Bollenbach group for support and discussions; and K. Mitosch, M. Lukačišinová, G. Liti and A. de Luna for critical reading of our manuscript. This work was supported in part by an Austrian Science Fund (FWF) standalone grant P 27201-B22 (to T.B.), HFSP program Grant RGP0042/2013 (to T.B.), EU Marie Curie Career Integration Grant No. 303507, and German Research Foundation (DFG) Collaborative Research Centre (SFB) 1310 (to T.B.). A.E.-C. was supported by a Georg Forster fellowship from the Alexander von Humboldt Foundation.

**Author contributions** Conceptualization: M.L. and T.B. Investigation: M.L. and A.E.-C. Writing original draft: M.L. Writing (review and editing): M.L., A.E.-C. and T.B. Supervision: T.B. Funding acquisition: T.B.

**Funding** Open access funding provided by Universität zu Köln.

**Competing interests** The authors declare no competing interests.

**Additional information**
**Correspondence and requests for materials** should be addressed to Tobias Bollenbach.

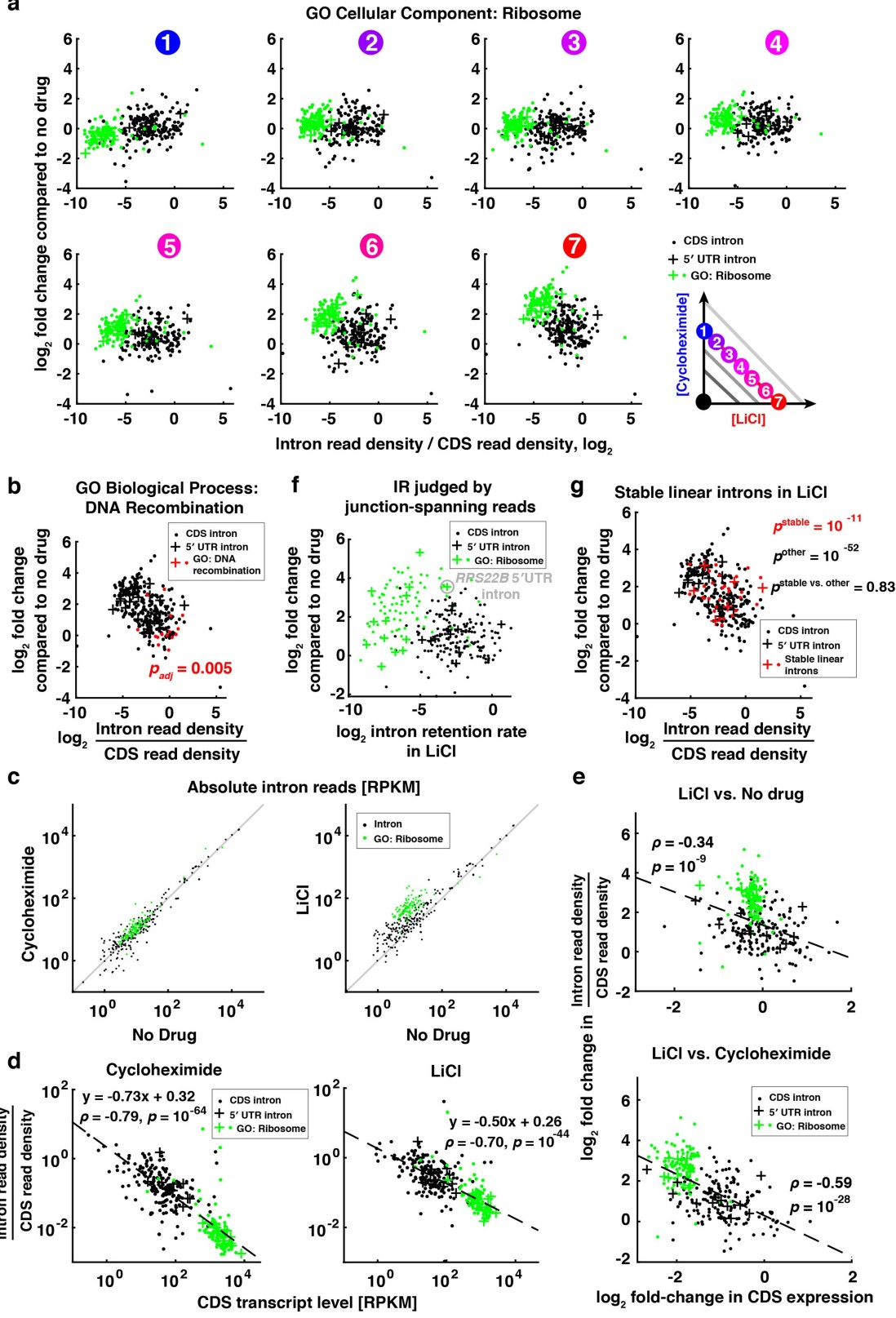

**Extended Data Fig. 1** | See next page for caption.

**Extended Data Fig. 1 | Introns in ribosomal proteins facilitate ribosomal protein response to LiCl stress. a**, Intron retention rates (as in Fig. 1c) along the 50% growth isobole of cycloheximide-LiCl drug combination. Introns contained in the genes belonging to the most significantly enriched GO Cellular Component with no offspring term as determined for intron retention increase in LiCl ('Ribosome') are highlighted in green. For quantification of intron retention rates from RNA-seq data and gene enrichment analysis refer to Methods. **b**, Intron retention rate in LiCl (as in panel 7 in **a**). The introns from genes belonging to the GO Biological Process term most significantly enriched in the lower end of the relative increase in intron retention ('DNA recombination') is displayed in red, with the hypergeometric test $p$-value adjusted for multiple hypothesis testing. **c**, Intron read density plots for all introns, comparing different conditions, show that intron reads in LiCl are elevated with respect to the entire transcriptome, not just with respect to their parent transcript. Introns in ribosomal genes are highlighted in green. Grey lines are visual guides for no change. RPKM – reads per kilobase of transcript per million mapped reads. **d**, Intron retention in cycloheximide (left) or LiCl (middle) compared to no drug for individual introns negatively correlates with the expression level of the corresponding RNA as gauged by the RNA-seq read counts in coding region (CDS). Linear model was fitted over all introns. **e**, Change in intron retention in LiCl negatively correlates with change in the expression of the corresponding CDS, both when compared to no drug (top) and to cycloheximide (bottom). Linear model was fitted over all introns. **f**, Analysis of intron retention using only the sequencing reads overlapping the exon-intron junction as the proxy for intron retention rate (Methods). The good agreement with the analysis being performed using all reads in the introns (panel A) supports that introns are being retained within the transcripts rather than just being spliced and non-degraded. **g**, Stable linear introns previously implicated in starvation response are increasingly retained in LiCl just like the rest of the introns. $p$-values are reported for one-sided $t$-testing of whether the two intron groups (stable, other) are unchanged in LiCl; two-sided $t$-test of whether the two groups behave the same in LiCl shows no significant difference. All data in this figure are based on sequencing ribo-depleted total RNA.

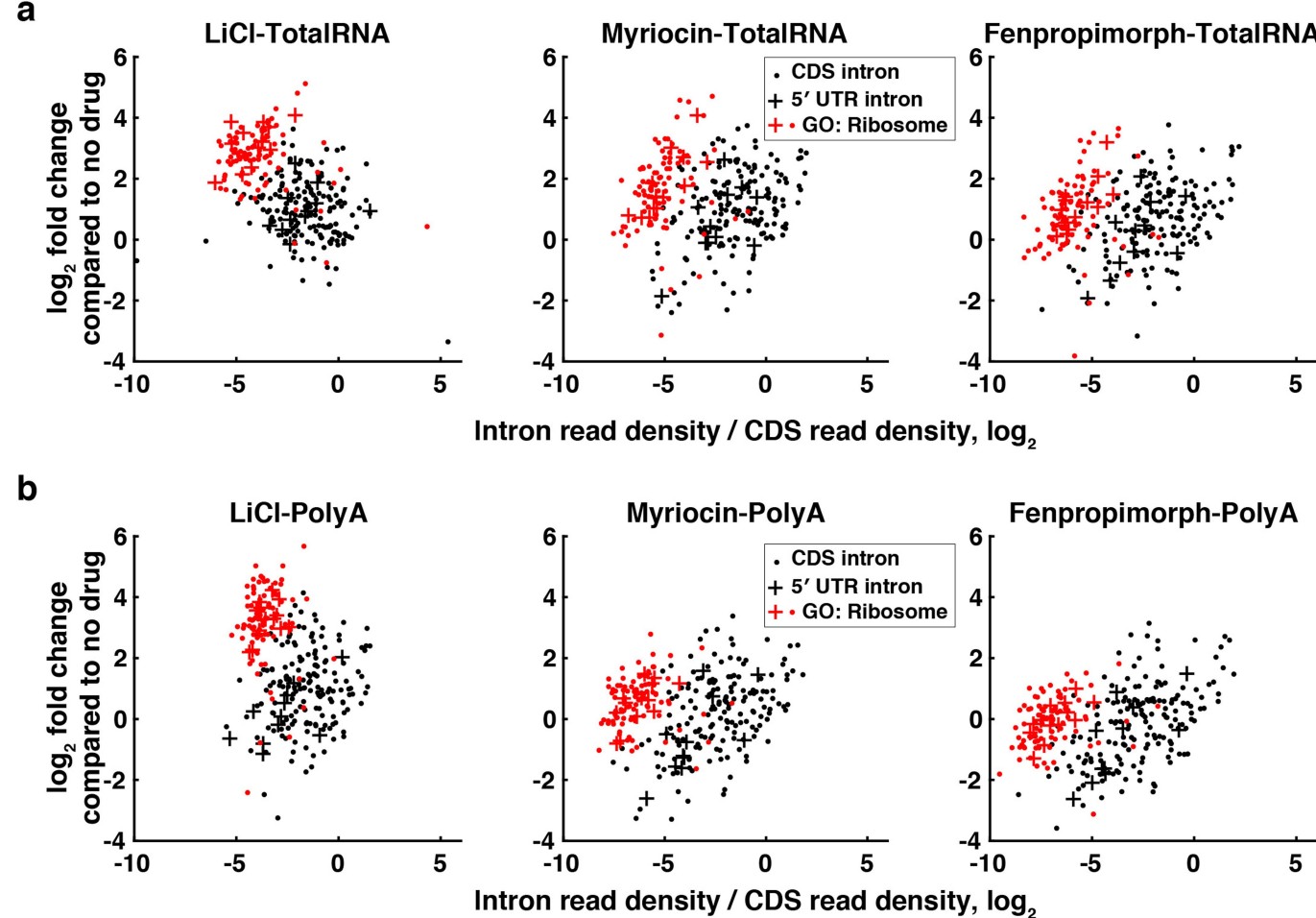

**Extended Data Fig. 2 | Sequencing of the poly-A RNA fraction confirms that LiCl induces intron retention. a**, Intron retention rate (intron read density / CDS read density) from total RNA-seq, as in Fig. 1c, for cultures grown in the presence of the drugs shown at their $IC_{50}$. **b**, As **a**, but for polyA-sequencing assay.

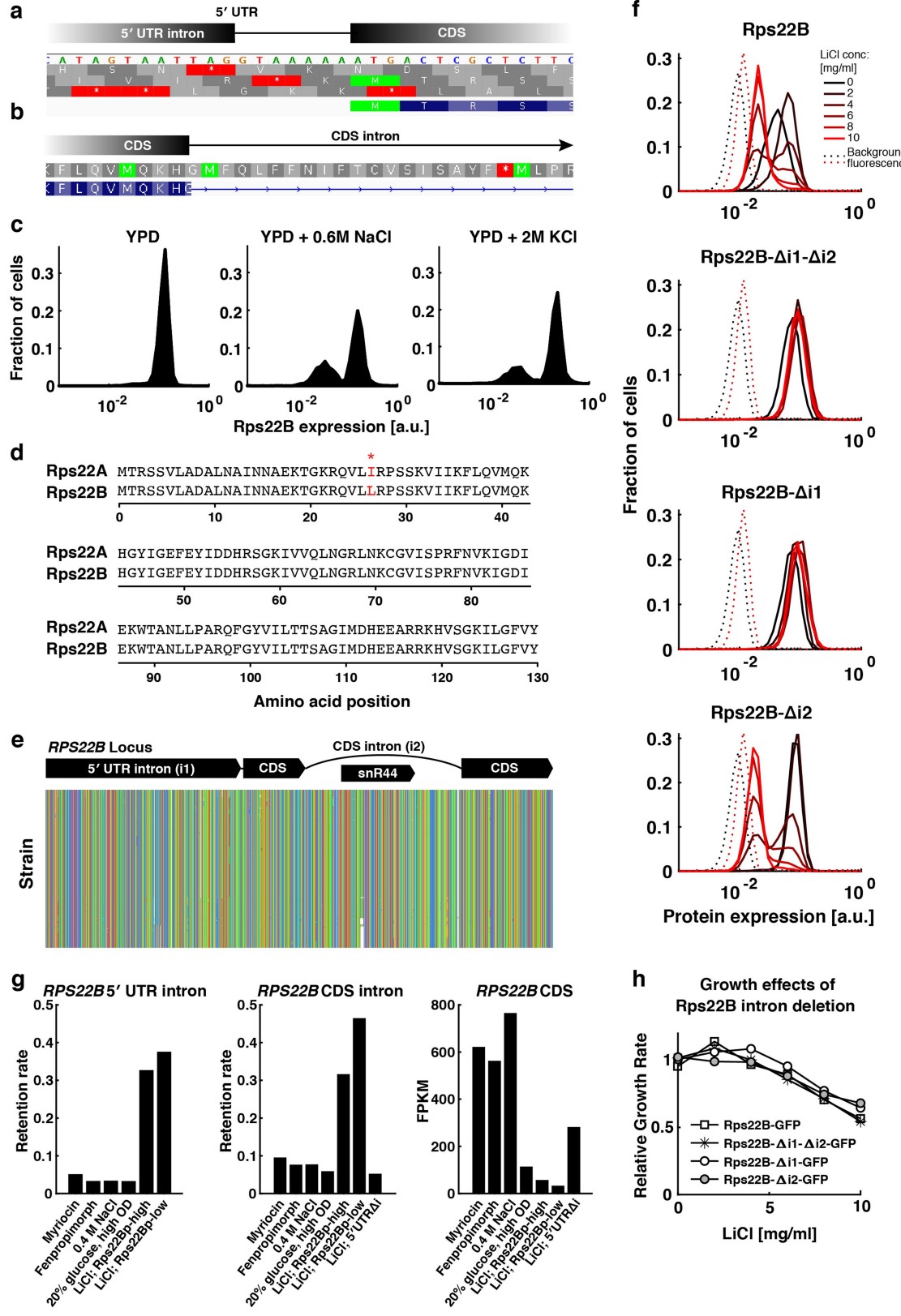

**Extended Data Fig. 3** | See next page for caption.

**Extended Data Fig. 3 | Osmotic stress induces bimodal expression in Rps22B. a**, DNA sequence and translation reading frames around the 3' splice site of the 5' UTR *RPS22B* intron. Stop codons (red) are present in the vicinity in all three reading frames. **b**, Rps22B amino acid sequence and the corresponding translation reading frame at the 5' splice site of *RPS22B* CDS intron, showing the presence of early stop codon in the intron. **c**, Histograms of Rps22B–GFP expression in NaCl and KCl, respectively, measured by flow cytometry during the exponential growth phase after 16 h of incubation. **d**, Protein blast for Rps22A and Rps22B. Red asterisk denotes the single difference between the amino acid sequences. RefSeq entries NP_013471.1 and NP_012345.1 are shown for Rps22A and Rps22B proteins, respectively. **e**, Multiple sequence alignment of previously published 1011 natural *Saccharomyces cerevisiae* isolates (Methods) reveals a remarkable conservation of the Rps22B 5' UTR intron during within-species evolution. Rows represent individual isolates, coloured columns represent individual bases in the Rps22B genomic locus (green=adenine, red=guanine, orange=cytosine, blue=thymine). **f**, Histograms of flow cytometry measurements of strains with GFP-tagged Rps22B, with seamless deletion of either or both introns in the presence of LiCl at different concentrations (*cf*. Fig. 2b). Δi1 denotes deletion of the 5' UTR intron, Δi2 deletion of the CDS intron. **g**, Intron retention (intron read density / CDS read density) and transcript level (FPKM) for *RPS22B* measured by RNA-seq of the polyA fraction in various conditions. Myricoin, fenpropimorph and LiCl were used at their respective $IC_{50}$. For details on sorting of the Rps22B high and low subpopulations, see Fig. 3g and Methods. FPKM – fragments per kilobase per million reads, i.e. read density normalized to the number of reads in the sequencing run. **h**, Changes in growth rate owing to intron deletion in Rps22B–GFP reporter strain cannot account for the observed differences in Rps22B expression pattern. Growth rates for the different strains are shown relative to the mean of their growth rates in the absence of drug.

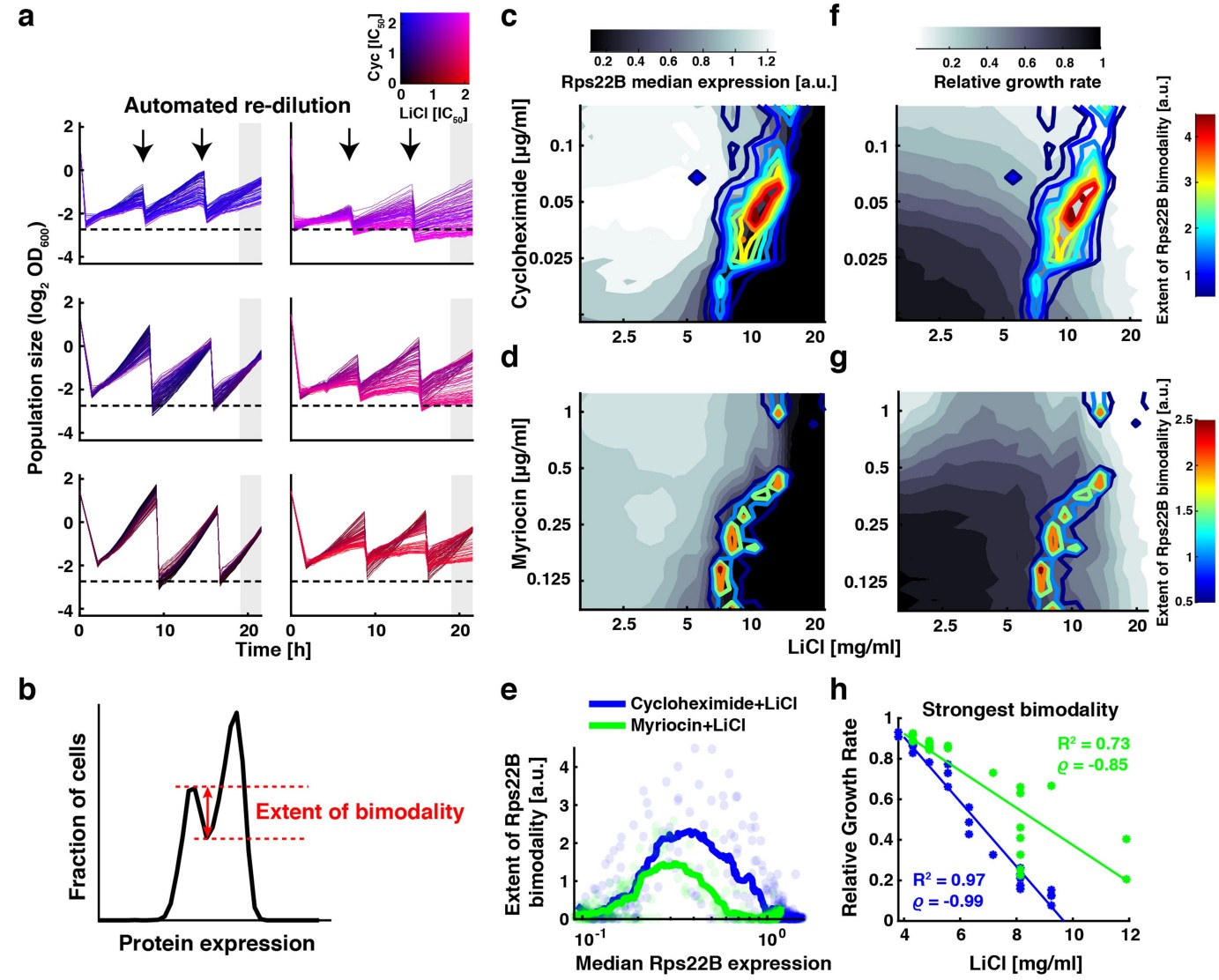

**Extended Data Fig. 4 | Rps22B protein expression is consistent with a bistable regulatory loop. a**, Growth curves from the automated re-inoculation setup of cultures growing in a detailed 2-drug gradient of LiCl and cycloheximide, spread over 576 wells of six microtitre plates. Each panel shows growth curves for cultures on one 96-well plate. Drops in $OD_{600\,nm}$ are due to automated re-dilution of growing cultures to an arbitrary target OD (dashed line). Shaded area denotes measurements that were used to determine the exponential growth rates shown in (**f**) and (**g**). **b**, Schematic showing the ad hoc definition of the quantity used to analyse the Rps22B bimodal expression pattern. The extent of bimodality is defined here as the y-axis distance between the trough and the lower of the two peaks on the histogram of single-cell protein expression, with bin size kept consistent throughout the analysis (Methods). **c**, **d**, The extent of Rps22B bimodality (coloured lines), is overlaid on Rps22B median expression (greyscale) in a 2-drug gradient of LiCl and cycloheximide (**c**) or myriocin (**d**). **e**, Rps22B protein bimodality as a function of median Rps22B protein expression for all wells in the 2-drug gradient (dots) is shown along with a running average with a window of 30 data points (lines). Rps22B bimodality peaks at a certain level of median Rps22B expression, rather than at a certain growth rate or LiCl concentration. **f**, **g**, The extent of Rps22B bimodality (coloured lines) is overlaid on growth rate (greyscale) in a 2D gradient of LiCl and cycloheximide (**f**) or myriocin (**g**). **h**, For each concentration of cycloheximide (blue) or myriocin (green), the LiCl concentration that elicited maximum Rps22B protein bimodality is plotted (x-axis), with the corresponding growth rate of the culture (y-axis). The strength of LiCl stress needed to bring the level of Rps22B to the putative unstable fixed point is inversely dependent on the growth rate, suggesting that the Rps22B switch is quantitatively attuned to specific growth conditions. $\varrho$ – Pearson correlation coefficient. See Extended Data Fig. 5 for Rps22B single-cell gene expression data for individual wells.

**a**

## 2-drug gradient of LiCl and cycloheximide

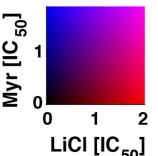

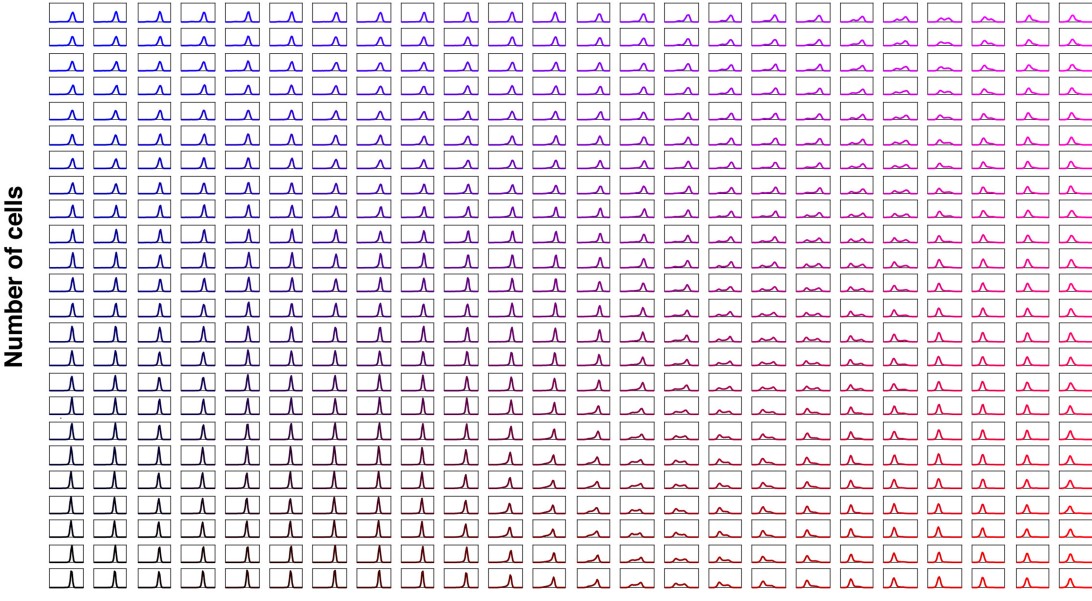

**Number of cells**

Rps22B-GFP single-cell intensity [a.u.]

**b**

## 2-drug gradient of LiCl and myriocin

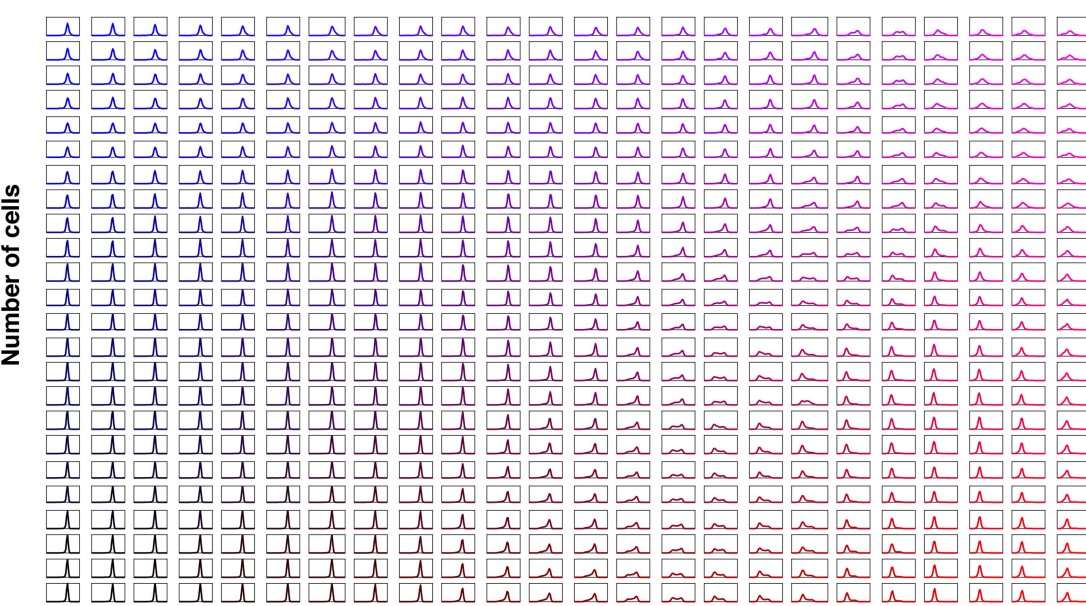

**Number of cells**

Rps22B-GFP single-cell intensity [a.u.]

**Extended Data Fig. 5 | Rps22B single-cell protein expression in a detailed two-drug gradient. a**, Histograms of flow cytometry measurements of strains with GFP-tagged Rps22B for 576 cultures with graded concentrations of LiCl and cycloheximide, incubated by automated re-inoculation protocol (Methods). **b**, As **a**, but for LiCl and myriocin.

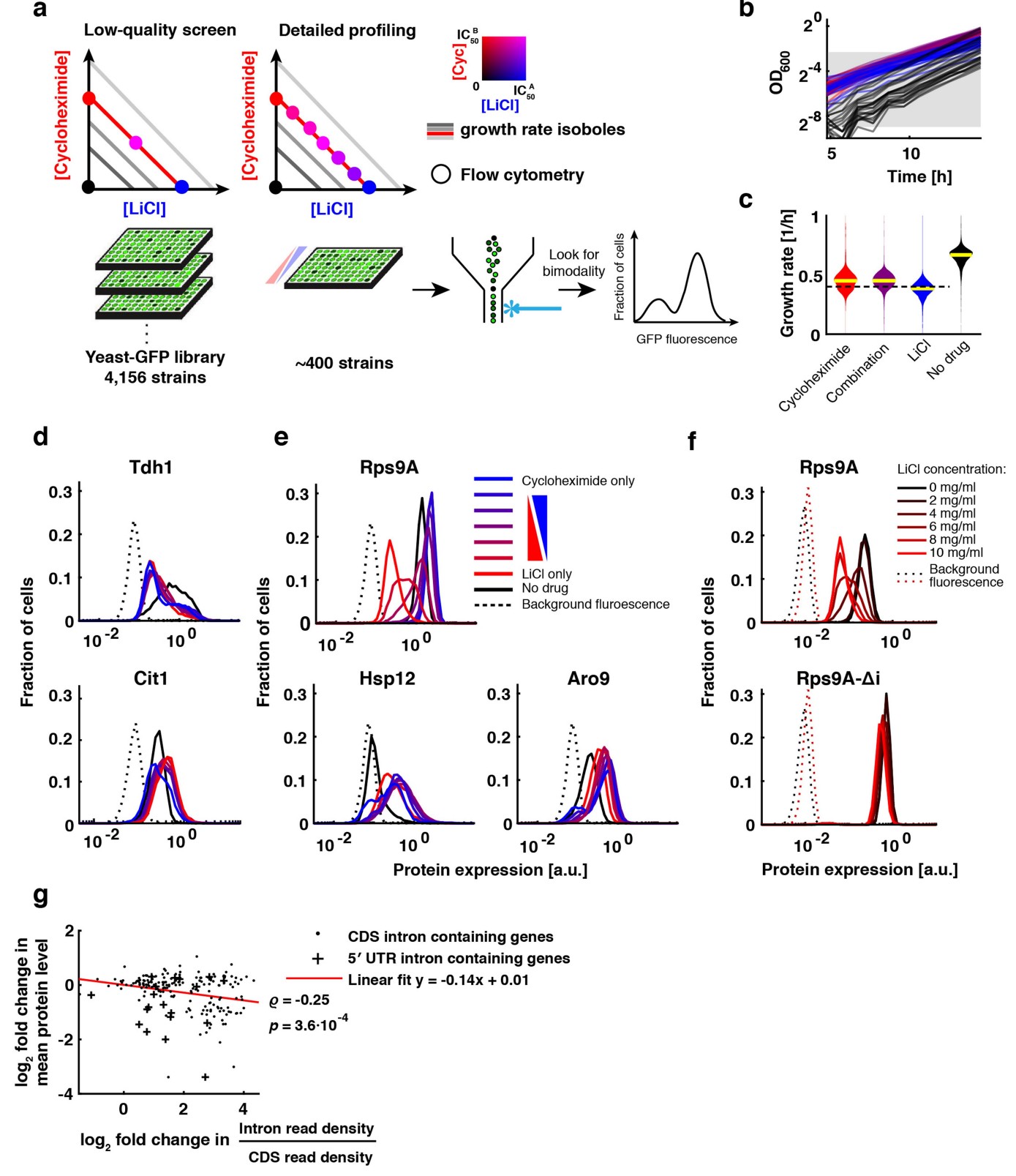

**Extended Data Fig. 6** | See next page for caption.

**Extended Data Fig. 6 | Single-cell isogrowth profiling uncovers drug-induced cellular states. a**, Schematic showing the experimental strategy for single-cell isogrowth profiling. Using flow cytometry, the entire yeast protein–GFP library was screened in four conditions: no drug, each of the two drugs alone at 40% inhibition, and an equipotent combination of drugs at the same total inhibition. For about 10% of genes which showed an unusual expression pattern, the corresponding strains were restreaked and subjected to a more detailed screen in 8 conditions. **b**, Example growth curves of one 96-well plate with 24 strains each in 4 conditions. Note that the inoculum of the no-drug condition (black) was decreased to achieve comparable final cell density at the time of collection. The shaded area shows the OD range used for fitting of growth rates. **c**, Violin plot showing the distribution of growth rates for the initial screen over the entire library. The dashed line indicates 40% growth inhibition. **d**, **e**, Single-cell GFP intensity histograms of protein–GFP fusion strains. Proteins with bimodal expression induced by growth inhibition by both LiCl and cycloheximide (**d**) or specifically by either drug (**e**) are shown. **f**, Intron deletion in Rps9A abolishes the LiCl-induced bimodality. **g**, Change in intron retention in LiCl compared to equal inhibition by cycloheximide as judged by RNA-seq is negatively correlated with the change in mean protein level as determined by flow cytometry. $\varrho$ – Pearson correlation coefficient; $p$ value was determined by two-sided permutation test.

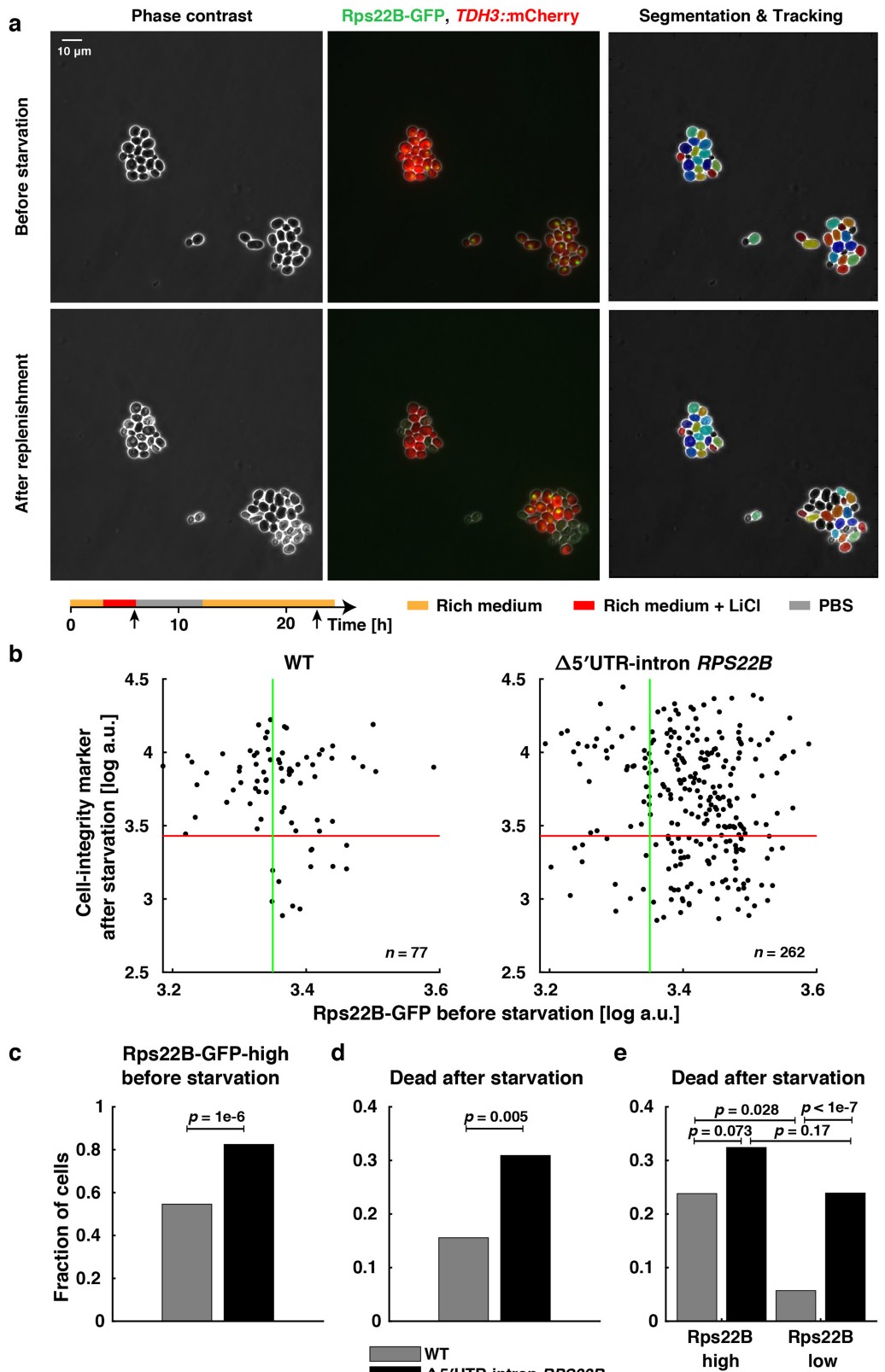

**Extended Data Fig. 7** | See next page for caption.

**Extended Data Fig. 7 | 5′ UTR intron in the *RPS22B* gene is necessary for the induction of phenotypic heterogeneity by LiCl. a**, Example micrographs of 5′ UTR intron-deleted Rps22B–GFP strain just before 6 h starvation in PBS (top row) and 10 h after rich medium replenishment (bottom row). Experimental setup is shown in the schematic below the micrographs; vertical arrows indicate timepoints at which the micrographs shown were taken. Scale bar = 10 μm. See Supplementary Videos 3, 4. **b**, Quantification of mCherry used as a cell integrity marker (cells lose mCherry signal after lysis) and GFP signal (Methods) in the micrographs from the timepoints in (**a**). **c**, 5′ UTR intron deletion in *RPS22B* increases the fraction of GFP-positive cells as defined by the green lines in (**b**). **d**, Same as Fig. 3k: Fraction of dead cells after starvation for WT (grey) and 5′ UTR intron-deletion mutant (black). 5′ UTR intron deletion in *RPS22B* increases the fraction of cells that do not survive the starvation stress, determined based on mCherry expression. The mCherry threshold for dead cells shown in (**b**) was chosen to be lower than mCherry expression of any cell before the starvation as well as lower than that of any cell after starvation that resumed growth. **e**, As (**d**), but showing Rps22B-high and -low subpopulations separately. The 5′ UTR intron deletion in *RPS22B* partially abolishes the coupling of the phenotypic heterogeneity in cell death to Rps22B protein expression heterogeneity. The *p*-values were determined using a one-sided permutation test.

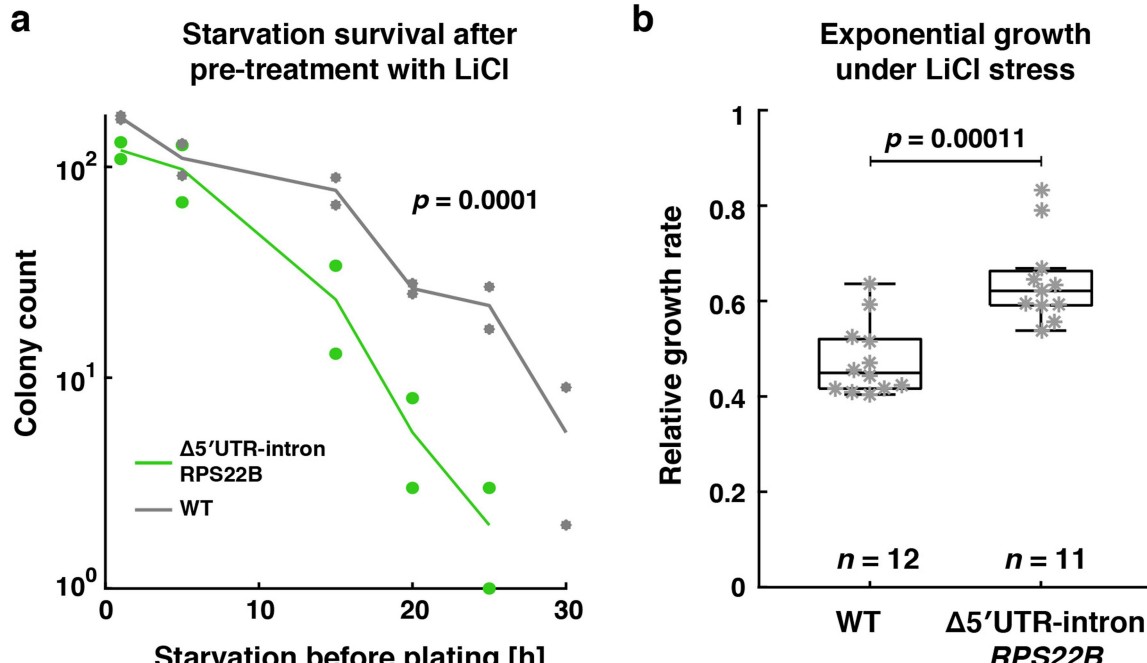

**a** Starvation survival after pre-treatment with LiCl

**b** Exponential growth under LiCl stress

**Extended Data Fig. 8 | Deletion of 5′ UTR intron in *RPS22B* confers phenotypic differences under starvation and osmotic stress. a,** Survival curves of the intron deletion mutant (green) and WT (grey), pre-cultured in YPD containing 9 mg/ml LiCl, when starved in PBS for varying duration before being plated on rich medium; pre-culturing the WT strain in LiCl at 9mg/ml results in a uniform population with low Rps22B expression (Fig. 2a), whereas deletion of the 5′ UTR intron results in a uniform population with high Rps22B expression (Fig. 2b). Significance was determined as in Fig. 3h. **b,** Exponential growth rates relative to uninhibited WT, of the 5′ UTR intron-deletion mutant and WT strain, in YPD containing 9 mg/ml LiCl. Two-sided *t*-test was used to determine significance. In each box, the central marker indicates the median, and the bottom and top edges of the box indicate the 25th and 75th percentiles, respectively. The whiskers extend to the most extreme data points not considered outliers.

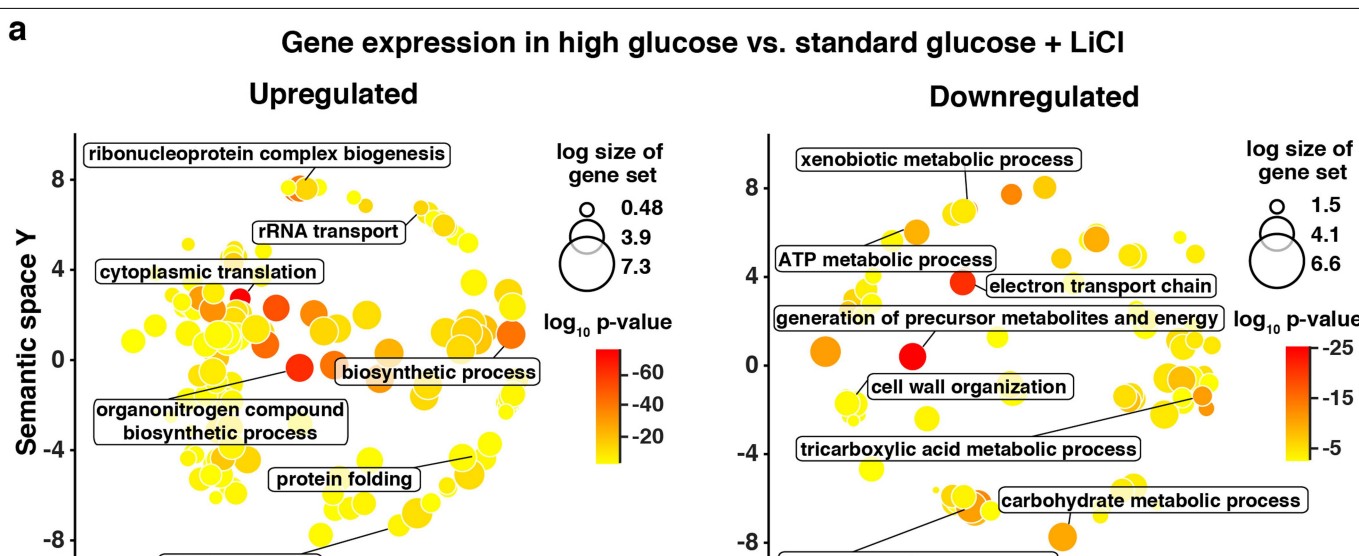

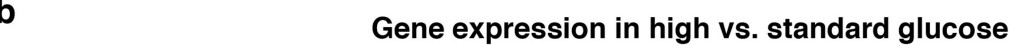

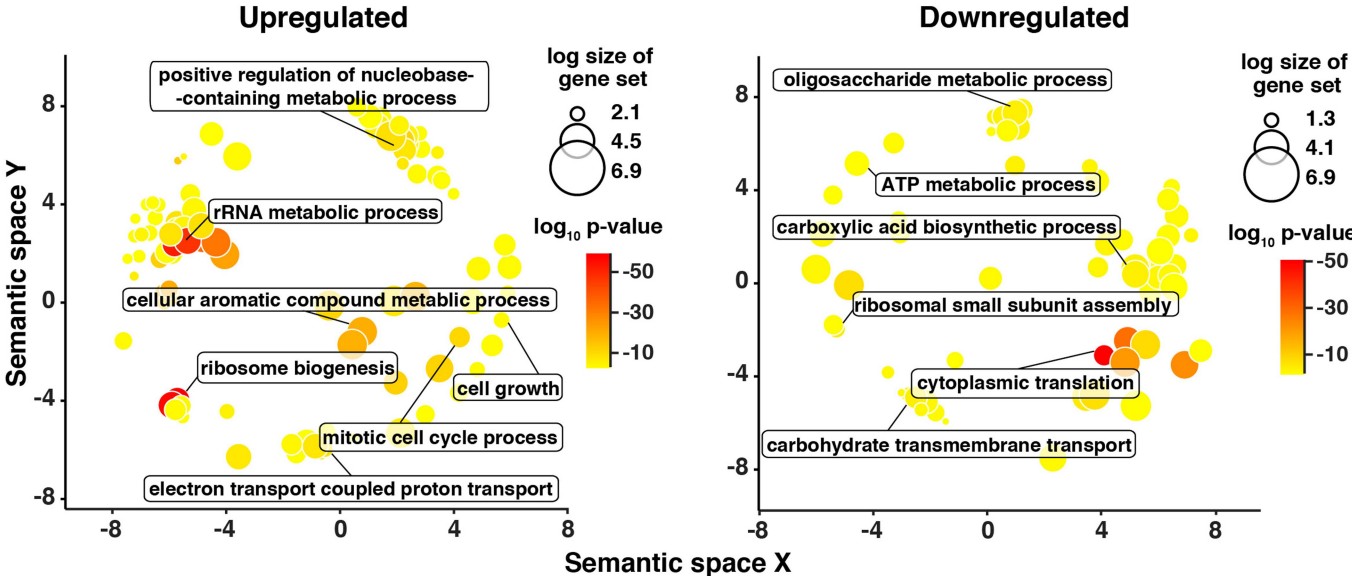

**Extended Data Fig. 9 | Functional enrichment analysis of gene expression changes in hyperosmotic glucose.** REVIGO visualizations (Methods) of GO biological process enrichment analysis of total RNA-seq reads from a yeast culture grown in 20% glucose to late exponential phase compared to a culture grown to mid-exponential phase in standard YPD containing $IC_{50}$ LiCl (**a**) or no drug (**b**). GO terms significantly upregulated (left) or downregulated (right) in high glucose are shown, along with GO term names for selected representatives of individual clusters.

**Extended Data Table 1 | Concentrations of drugs used**

**a.**

| Condition (as labelled in Extended Data Fig. 1) | Cycloheximide [µg/ml] | LiCl [mg/ml] | LiCl [mM] |
|---|---|---|---|
| 1 | 0.11 | 0 | 0 |
| 2 | 0.095 | 1.2 | 28 |
| 3 | 0.083 | 2.5 | 59 |
| 4 | 0.069 | 4.1 | 97 |
| 5 | 0.049 | 5.9 | 139 |
| 6 | 0.026 | 7.7 | 182 |
| 7 | 0 | 8.5 | 201 |

**b.**

| Condition | Cycloheximide [µg/ml] | LiCl [mg/ml] | LiCl [mM] |
|---|---|---|---|
| Cycloheximide only | 0.10 | 0 | 0 |
| Cycloheximide + LiCl | 0.050 | 6.3 | 149 |
| LiCl only | 0 | 13.5 | 318 |
| YPD only | 0 | 0 | 0 |

**c.**

| Condition | Cycloheximide [µg/ml] | LiCl [mg/ml] | LiCl [mM] |
|---|---|---|---|
| 1 Cycloheximide only | 0.10 | 0 | 0 |
| 2 | 0.083 | 2.7 | 64 |
| 3 | 0.067 | 4.7 | 111 |
| 4 | 0.052 | 6.7 | 159 |
| 5 | 0.036 | 8.8 | 206 |
| 6 | 0.021 | 10.8 | 254 |
| 7 LiCl only | 0 | 13.5 | 318 |
| 8 No drug control | 0 | 0 | 0 |

**a**, For the RNA-isogrowth profiling. **b**, In the whole yeast protein–GFP library isogrowth screen. **c**, For the more detailed antiparallel gradient used to profile a subset of protein–GFP library strains. LiCl concentration is stated both in weight and molar concentration for an easier comparison to published literature.

# Reporting Summary

Nature Research wishes to improve the reproducibility of the work that we publish. This form provides structure for consistency and transparency in reporting. For further information on Nature Research policies, see our Editorial Policies and the Editorial Policy Checklist.

## Statistics

For all statistical analyses, confirm that the following items are present in the figure legend, table legend, main text, or Methods section.

| n/a | Confirmed | |
|-----|-----------|---|
| ☐ | ☒ | The exact sample size ($n$) for each experimental group/condition, given as a discrete number and unit of measurement |
| ☐ | ☒ | A statement on whether measurements were taken from distinct samples or whether the same sample was measured repeatedly |
| ☐ | ☒ | The statistical test(s) used AND whether they are one- or two-sided<br>*Only common tests should be described solely by name; describe more complex techniques in the Methods section.* |
| ☒ | ☐ | A description of all covariates tested |
| ☐ | ☒ | A description of any assumptions or corrections, such as tests of normality and adjustment for multiple comparisons |
| ☐ | ☒ | A full description of the statistical parameters including central tendency (e.g. means) or other basic estimates (e.g. regression coefficient) AND variation (e.g. standard deviation) or associated estimates of uncertainty (e.g. confidence intervals) |
| ☐ | ☒ | For null hypothesis testing, the test statistic (e.g. $F$, $t$, $r$) with confidence intervals, effect sizes, degrees of freedom and $P$ value noted<br>*Give P values as exact values whenever suitable.* |
| ☒ | ☐ | For Bayesian analysis, information on the choice of priors and Markov chain Monte Carlo settings |
| ☒ | ☐ | For hierarchical and complex designs, identification of the appropriate level for tests and full reporting of outcomes |
| ☐ | ☒ | Estimates of effect sizes (e.g. Cohen's $d$, Pearson's $r$), indicating how they were calculated |

*Our web collection on statistics for biologists contains articles on many of the points above.*

## Software and code

Policy information about availability of computer code

| Data collection | FACS Diva, Tecan FreedomEvo, NIS elements, CellASIC ONIX2 System software |
|---|---|
| Data analysis | TopHat (v2.1.1), featureCounts v. 1.6.0 and 1.6.2., GOrilla (accessed online 2016-2022), Matlab R2012b-R2019b, Clustal Omega (accessed online, 2021), MSA-BIOJS (accessed online 2021), CellStar Matlab plugin (v. 1.0.1), Integrative Genomics Viewer (v. 2.3.80 and 2.9.4), HISAT2 (v. 2.1.0), Revigo (accessed online 2021), YeaZ ( v. 2021-08-16). |

For manuscripts utilizing custom algorithms or software that are central to the research but not yet described in published literature, software must be made available to editors and reviewers. We strongly encourage code deposition in a community repository (e.g. GitHub). See the Nature Research guidelines for submitting code & software for further information.

## Data

Policy information about availability of data

All manuscripts must include a data availability statement. This statement should provide the following information, where applicable:

- Accession codes, unique identifiers, or web links for publicly available datasets
- A list of figures that have associated raw data
- A description of any restrictions on data availability

The RNA sequencing data are available in Gene Expression Omnibus under accession no. GSE155060 and GSE197174. The processed flow cytometry data from isogrowth profiling are available as Supplementary Table 1 and 2 accompanying this manuscript. S. cerevisiae R64-2 reference genome and annotation was downloaded from National Center for Biotechnology Information, assembly ref. GCF_000146045.2.

# Field-specific reporting

Please select the one below that is the best fit for your research. If you are not sure, read the appropriate sections before making your selection.

☒ Life sciences          ☐ Behavioural & social sciences          ☐ Ecological, evolutionary & environmental sciences

For a reference copy of the document with all sections, see nature.com/documents/nr-reporting-summary-flat.pdf

# Life sciences study design

All studies must disclose on these points even when the disclosure is negative.

| | |
|---|---|
| Sample size | Sample sizes were determined by technical constraints. For the flow cytometry measurements, each sample was measured for 10s or until 10000 events were measured, whichever was earlier, in order to achieve feasible timescale of measuring the entire collection. For the microscopy time-lapse measurements, a few fields of view (≤13) were imaged, as imaging more fields of view usually led to the loss of focus and thus was not technically feasible. |
| Data exclusions | The strain Smi1 appeared bimodal in the library screen; however, its identity could not be confirmed by PCR and thus was excluded from reporting. For growth rate measurements of sorted cells, growth rates higher than 0.85/h, which resulted from bacterial contaminations, were disregarded. |
| Replication | Rps22B protein bimodality in LiCl was replicated independently by two researchers, at least five times, using two different yeast strains; all attempts at replication were successful. Rps22B protein bimodality in NaCl and KCl was determined twice; all attempts at replication were successful. The loss of bimodality in the RPS22B 5' UTR intron deletion strain was replicated three times; all replication attempts were successful. The bimodal expression of the RPS22B 5' UTR intron-GFP fusion in LiCl was observed twice; all attempts were successful. The Rps22B bimodality on the entry to stationary phase after growth in high glucose was reproduced twice; the exact timing of observing maximum bimodality varied, so multiple time points were measured as described in the Methods, Rps22 bimodality in other osmotic stresses and in high glucose. All attempts at replication using the incubation method described were successful; one attempt at replication using a different microtiter plate shaker (Titramax 1000, Heidolph) resulted in less pronounced heterogeneity, likely due to a different degree of aeration/shaking. Intron retention determined by sequencing was confirmed independently by two researchers, once by each; all attempts at replication were successful. Time-lapse microscopy phenotype of Rps22B-high and low cells was replicated twice in slightly modified setups and with independent strains as detailed in Fig. 3 and Extended Data Fig. 7, all attempts at replication were successful; for the experiment shown in Extended Data Fig. 7, two starvation conditions, in which the length of the starvation period was varied, were tested in order to achieve an intermediate number of cell deaths within the timeframe of the experiment as assessed visually and the chosen condition was then quantified. The whole yeast-GFP library isogrowth scan was performed once due to the scale of experimental effort; about 10% of strains that did not show clear unimodal expression on visual inspection of the expression histograms were restreaked to exclude the possibility of contaminations and the measurement was repeated once on a more detailed antiparallel gradient as detailed in the manuscript; the bimodality of Rps9A and Aro9 was additionally replicated once in a detailed discretised 2D drug concentration gradient similar to the one shown in the manuscript for Rps22B. To determine the phenotypic effect of Rps22B expression level, cells were sorted once and survival and growth assays performed in two and 24 replicates, respectively; the phenotypic effect was independently confirmed by other methods (time-lapse microscopy and phenotypic assays using the intron-deletion mutant in Extended Data Fig. 8). |
| Randomization | There were no experimental groups that would warrant randomization. |
| Blinding | There were no experimental groups that would warrant blinding. |

# Reporting for specific materials, systems and methods

We require information from authors about some types of materials, experimental systems and methods used in many studies. Here, indicate whether each material, system or method listed is relevant to your study. If you are not sure if a list item applies to your research, read the appropriate section before selecting a response.

## Materials & experimental systems

| n/a | Involved in the study |
|---|---|
| ☒ | Antibodies |
| ☒ | Eukaryotic cell lines |
| ☒ | Palaeontology and archaeology |
| ☒ | Animals and other organisms |
| ☒ | Human research participants |
| ☒ | Clinical data |
| ☒ | Dual use research of concern |

## Methods

| n/a | Involved in the study |
|---|---|
| ☒ | ChIP-seq |
| ☐ | ☒ Flow cytometry |
| ☒ | MRI-based neuroimaging |

# Flow Cytometry

## Plots

Confirm that:

☒ The axis labels state the marker and fluorochrome used (e.g. CD4-FITC).

☒ The axis scales are clearly visible. Include numbers along axes only for bottom left plot of group (a 'group' is an analysis of identical markers).

☒ All plots are contour plots with outliers or pseudocolor plots.

☒ A numerical value for number of cells or percentage (with statistics) is provided.

## Methodology

Sample preparation
> For flow cytometry, the yeast cells in 96-well microtitre plates were twice: centrifuged at 1050 g for 3.5 min and resuspended in ice-cold Tris-EDTA buffer by vigorous shaking at 1000 rpm on a Titramax shaker for 30 s. After another centrifugation at 1050 g for 3.5 min, the cells were resuspended in 80 μl of Tris-EDTA and immediately stored at -80°C. On the day of the flow cytometry measurement, the plates were thawed on ice for ~3 hrs and kept on ice until the measurement. In the course of study, an alternative simplified protocol was introduced (cf. Methods), where the growing cultures were measured by reading OD, an automatic dilution with Tris-EDTA to a target OD was performed, and the flow cytometry reading was conducted immediately afterwards.
> For the FACS sorting, the Rps22B-GFP strain with constitutive mCherry expression was inoculated into YPD and incubated at 30°C for 16 hours. The overnight culture was diluted 20-fold into 4.5 mg/ml LiCl in YPD, incubated for 6 h and washed twice with PBS. Positive mCherry cells were sorted into low-GFP and high-GFP populations with a Becton Dickinson INFLUX cell sorter.

Instrument
> BD FACS Canto II, BD INFLUX

Software
> BDFACSDiva, Matlab

Cell population abundance
> 64.64 (RPS22B-GFP-low) and 35.31 (RPS22B -GFP-high)

Gating strategy
> mCherry positive (constituitive expression, cytoplasmic marker) and then either Rps22GFP high or low.

☒ Tick this box to confirm that a figure exemplifying the gating strategy is provided in the Supplementary Information.

