## [Peer Review File · Nature]

Manuscript Title: Intron-mediated induction of phenotypic heterogeneity

Reviewer Comments & Author Rebuttals

Reviewer Reports on the Initial Version:

Referees' comments:

Referee #1 (Remarks to the Author):

Lukacisin et al describe a series of interesting experiments in which they demonstrate that the 5' UTR-intron of the ribosomal protein RPS22B confers heterogeneity in expression in a particular set of conditions. When the intron is present in the transcript two subpopulations of cells are present in starved cells. One in which the protein expression level is low allowing the cells to survive prolonged starvation and one in which the protein expression is high allowing the cells to resume growth rapidly following prolonged starvation. They argue that this provides a means of phenotypic diversification in nutrient stress conditions.

This is an interesting study. It provides insight into the unresolved question regarding the retention of gene duplicates and extends recent studies on the implications of differential regulation of gene duplicates in yeast. However, the extent to which this phenomenon generalizes to other transcripts and other species is unclear. Moreover, there is no insight provided into the molecular mechanism that underlies the reported behavior.

The authors should consider the following point prior to publication.

-Stress induces a decrease in the total amount of transcript. Therefore, what the authors are measuring is an increase in the relative abundance of transcripts that retain transcripts, but not necessarily an increase in the total number. Is this due to a global reduction in the activity of the splicing machinery?

-How does the increase in intron retention relate to the reported increase in linear introns in starvation condition in yeast? Are they connected or totally independent phenomena?

-It is never clearly explained whether the 5' UTR-intron is in frame and/or contains a start codon.

-For all other transcripts that retain introns, what is the impact on the sequence of the encoded peptide? How many would remain in frame?

-Are the retained introns translated? This could be assessed using ribosome profiling.

-The comparison to cycloheximide is somewhat confounded by the fact that cycloheximide stops or

slows ribosomes on transcripts thereby potentially altering RNA decay kinetics.

-Is the CDS intron that is contained in the RPS22B locus also retained? If not, what is the cause of this difference in retention rates within the same transcript?

-Does fusion of the 5' intron to an unrelated transcript induce bimodal gene expression in the same condition or is this behavior unique to RPS22B?

-I understand the justification for using a high glucose concentration, however, it remains unclear what signal the cell is sensing here. In such a high concentration of glucose is cell growth limited by some other essential nutrient (e.g. nitrogen) and therefore is the signal that underlies the response?

-Can the transcript be engineered so that 100% of the transcripts retain the intron perhaps by altering the splice site sequence? This would allow testing of whether the bimodal response is due to the intron itself or if the bimodal expression is a result of having both intron and intronless transcripts.

Referee #2 (Remarks to the Author):

Multiple theories exist for why budding yeasts like *Saccharomyces cerevisiae* have retained paralogous ribosomal protein pairs for millions of years after an ancestral whole-genome duplication. Several pairs include one paralog without introns and another with introns that are conserved across species, hinting that differential regulation could provide cellular diversity and differential ribosome functions. In this work, the authors investigate stress-dependent intron retention in the RPS22B transcript and its effects on protein expression in yeast. They show that high doses of LiCl stress reduce splicing of the transcript to modulate protein abundance. Interestingly, intermediate doses of LiCl cause cell-to-cell variability in Rps22b protein abundance, and this heterogeneity requires the 5' intron of the transcript. The authors go on to show that intermediate doses of LiCl produce culture heterogeneity, such that some cells in the culture are better able to survive long-term starvation and others instead thrive during regrowth after nutrient repletion. This type of bet hedging is an important strategy in microbes that have evolved to compete in natural settings with unpredictable environmental shifts.

The manuscript is well presented (although a little over-sold in places in my opinion) and clear, and this is an interesting topic. However, while the authors provide a nice example of how splicing can regulate protein abundance during stress (something that is already known and should be made clearer here), they do not show that heterogeneity in intron retention is causing the phenotypic variation. It is already known that yeast cells vary in their metabolism in a way that produces cell-to-cell heterogeneity in how cells respond to starvation (e.g. recent paper by Bagamery et al. in *Current Biology*). Most likely, the heterogeneity arises from some cellular signal above RPS22B splicing and that variable signal causes the starvation phenotypes – in other words, RPS22B intron retention is likely an output of this heterogeneity rather than the cause. I would be otherwise convinced if the authors showed that the mutant lacking the 5' intron, which lacks the Rps22b protein heterogeneity, lost the heterogeneity in starvation responses.

Without a causal link between intron retention and differential starvation response, I think this is nice paper but perhaps not enough sufficiently novel insights to warrant publication in Nature.

Additional comments below.

1) The growth regime should be made clearer. Personally, I don't see the rationale of this method and really the main result is that LiCl alone affects intron retention. At any rate, a clearer description that the "gradient" is not physical but rather a series of cultures with different doses would be useful.

2) In Fig 2D, what does "extent" of bimodality refer to? The median Rps22b expression on the x-axis is described as "for ... wells" but it seems to me that this should be the median abundance per cell? High heterogeneity can give an intermediate culture-level abundance if half the cells have high protein production and half have low.

3) The authors do a nice job showing the heterogeneity in protein levels. I was left wondering if there is heterogeneity in intron retention in the same cultures (as opposed to low levels of splicing in all cells and heterogeneity comes from mass-action differences in translation initiation). This could be addressed with single-molecule FISH against the intron in individual cells.

4) Please list LiCl concentrations in mM so as to more easily compare to other papers that cite concentration.

Referee #3 (Remarks to the Author):

Intron-mediated induction of phenotypic heterogeneity by Lukacisin et al. describes a new phenomenon where stress causes the retention of an intron in a ribosomal protein transcript and thus presumably affects ribosome level. They go on to show that this bimodality can lead to a bet-hedging strategy that has a fitness trade-off depending on environmental conditions. Overall, this is an interesting study with important implications. To me, the critical question is whether this phenomenon is real. While I am largely convinced, I have a couple reservations and experiments I would like to see that would help convince me that the regulation isn't a complex artifact.

1). Is this intron retention or more nascent transcription? In theory, S1C was meant to address this but it compared LiCl to no drug treatment which convolutes growth rate. I think the isogrowth profiling is critical and needs to be maintained here. S1C should be redone but compared to cycloheximide at the same growth condition.

Relatedly, given the importance of really solidly showing the phenomenon is real, I think the paper would be greatly strengthened by quantitation of the intron/exon ratio in the polyA fragment. This would more directly detect if this was intron retention or just higher transcriptional rates or less stable mature transcripts. This could be done by sequencing or qPCR.

2) Figure 1C and 1D were a bit tough to understand at first. It wasn't clear at first that all introns were

plotted. Given the density I wouldn't be able to tell if the '+' 'x' symbols were non-uniformly distributed. The x-axis was a bit confusing and should probably be labeled as Intron density/exon density. I actually found S1A super helpful and almost feel that part of that data should be incorporated in the main figure. I really wanted to see a direct plot of intron retention versus transcript level. Looking at the plots in S1A made me worry that this could just be more nascent transcripts.

3) The mutant data around intron deletions are strong. Some direct quantification of transcript level in Fig 2 would probably help make this stronger along.

4) Where do the long and short starvation times come from? The difference doesn't seem that large. What is the time frame of intron retention during starvation? Shouldn't that be what sets the short and long times. A time course of the change in intron retention would be nice and then you could use that to let the cells tell you what is the relevant time scale (unless there is some good exterior justification for the chosen starvation times)

5) The difference in budding in 3F doesn't look that overwhelming given the variance. Did you check other phenomenon? Was this multiple hypothesis corrected? I worry that this might not be significant. Maybe plotting 3F as a beeswarm or similar plot so we can see all the data points would help some.

Minor

1) The transition to the bistability was a bit abrupt. I would recommend some statement around line 108 that more clearly states how the 5-intron retention would affect level, e.g. leads to a frame shift that leads to an early stop. This would then make it clear that looking at single cell protein levels would be a natural way to follow the phenomenon in multiple conditions.

2) Should 160-176 come before 142-157? I ended up re-reading it in that order when I went through it a second time.

Author Rebuttals to Initial Comments:

Point-by-point response

Reviewer #1:

Lukacisin et al describe a series of interesting experiments in which they demonstrate that the 5' UTR-intron of the ribosomal protein RPS22B confers heterogeneity in expression in a particular set of conditions. When the intron is present in the transcript two subpopulations of cells are present in starved cells. One in which the protein expression level is low allowing the cells to survive prolonged starvation and one in which the protein expression is high allowing the cells to resume growth rapidly following prolonged starvation. They argue that this provides a means of phenotypic diversification in nutrient stress conditions.

This is an interesting study. It provides insight into the unresolved question regarding the retention of gene duplicates and extends recent studies on the implications of differential regulation of gene duplicates in yeast. However, the extent to which this phenomenon generalizes to other transcripts and other species is unclear. Moreover, there is no insight provided into the molecular mechanism that underlies the reported behavior.

The authors should consider the following point prior to publication.

Stress induces a decrease in the total amount of transcript. Therefore, what the authors are measuring is an increase in the relative abundance of transcripts that retain introns, but not necessarily an increase in the total number. Is this due to a global reduction in the activity of the splicing machinery?

We thank the reviewer for raising this point, as it illustrates the utility of our isogrowth profiling approach. It is correct that we measure the relative abundance of transcripts that retain introns. However, the observed increase in intron retention is certainly not due to a global reduction in the activity of the splicing machinery that could occur generally under stress because it is specific to LiCl and does not occur for other drugs. To corroborate this point, we performed a new analysis of the number of reads related to introns (normalized to the entire transcriptome), which shows that they increase specifically in LiCl (and specifically for ribosomal proteins, new **Fig. S1C**). No such increase occurs in cycloheximide at a concentration causing the same growth inhibition (**Fig. S1C**). Similarly, no such increase in intron retention was observed in response to two additional unrelated drugs, namely the ergosterol synthesis inhibitor fenpropimorph and the sphingolipid synthesis inhibitor myriocin (new **Fig. S2**). Thus, we conclude that the intron retention we observed in LiCl is not a result of a decrease in the total amount of transcript due to a general stress response, but rather a specific response to LiCl. We now mention this explicitly in lines 90-92.

How does the increase in intron retention relate to the reported increase in linear introns in starvation condition in yeast? Are they connected or totally independent phenomena?

We agree that it is interesting to compare the intron retention we observe here to previously reported phenomena. Additional analysis showed that the behavior of the previously reported 34 stable linear introns (Morgan et al., 2019, PMID:30651636) is indistinguishable from that of the other introns under LiCl (two sample *t*-test *p*-value = 0.83, new **Fig. S1G**). This suggests that the phenomenon we report here is independent of that observed under starvation by Morgan et al., 2019. We now explain this point in lines 98-102 of the revised manuscript.

It is never clearly explained whether the 5' UTR-intron is in frame and/or contains a start codon.

The first intron in the *RPS22B* transcript is located in the 5' untranslated region (5' UTR), i.e. it is upstream of the annotated start codon and thus not translated and neither in-frame nor out-of-frame. We also checked that it is not feasible that retention of the 5' UTR intron in the *RPS22B* transcript creates an alternative start codon, since there is at least one stop codon just before the canonical start codon in each of the three possible reading frames. We concur that it is instructive to explain this clearly and have included this information in the main text (lines 122-124) as well as in the new **Fig. S3A**.

For all other transcripts that retain introns, what is the impact on the sequence of the encoded peptide? How many would remain in frame? Are the retained introns translated? This could be assessed using ribosome profiling.

This interesting topic has previously been explored in depth, including by ribosome profiling, see e.g. Zafirir et al., 2016, PMID:27260512; Behringer & Hall, 2016, PMID:27630196. In brief, premature stop codons are pervasive in the introns of *S. cerevisiae*, i.e. a common effect of retained introns (in the coding sequence) is partial translation resulting in a truncated protein as long as the mRNA is not removed by the nonsense-mediated decay pathway. Although our manuscript focuses on the effects of a 5'UTR intron, we agree that this information provides relevant context for our results and now mention it in the main text (line 123).

The comparison to cycloheximide is somewhat confounded by the fact that cycloheximide stops or slows ribosomes on transcripts thereby potentially altering RNA decay kinetics.

We agree with the reviewer that additional reference conditions are helpful to validate that the observed effect is not due to an unusual effect of cycloheximide. We have now included a comparison to cultures inhibited by two additional drugs with unrelated modes of action: myriocin (inhibitor of sphingolipid synthesis) and fenpropimorph (inhibitor of ergosterol synthesis). At their respective IC₅₀, no comparable increase in intron retention is observed for

either drug (new **Fig. S2**), confirming that the observed increased intron retention is specific for LiCl.

Is the CDS intron that is contained in the RPS22B locus also retained? If not, what is the cause of this difference in retention rates within the same Transcript?

This is an interesting question. The exact quantification of the CDS intron retention rate was previously confounded by the use of total RNA-seq (sequencing all RNA except for the depleted rRNAs) and the presence of a separately controlled small RNA (snR44) within the CDS intron. To resolve this issue, we have now performed polyA sequencing (which retains only mature mRNAs, hence no small RNAs) of the strain grown in LiCl at IC₅₀. The CDS intron is retained to a similar level as the 5' UTR intron (new **Fig. S3G**).

Does fusion of the 5' intron to an unrelated transcript induce bimodal gene expression in the same condition or is this behavior unique to RPS22B?

We thank the reviewer for raising this interesting issue, which can provide insights into the molecular mechanism underlying the observed bimodal gene expression. To address this point, we analyzed the effect of the 5' UTR of *RPS22B* fused to a GFP transcript. Notably, this construct exhibited bimodal expression in the presence of LiCl (new **Fig. 2C**), similar to that observed for *RPS22B*. Fusing the intron-deleted 5' UTR of *RPS22B* to GFP abolished this bimodality (new **Fig. 2C**). These results show that the 5' UTR of *RPS22B* is sufficient to elicit bimodal gene expression and corroborate that the 5' UTR intron is necessary for this effect. We have included these new results in the main text (lines 142-146).

I understand the justification for using a high glucose concentration, however, it remains unclear what signal the cell is sensing here. In such a high concentration of glucose is cell growth limited by some other essential nutrient (e.g. nitrogen) and therefore is the signal that underlies the response?

This is an interesting point but challenging to address. To explore the signals the cell is sensing and responding to, we have performed RNA sequencing of cultures grown at high glucose concentrations nearing stationary phase (where the depletion of limiting nutrients starts to have an effect) and compared their genome-wide gene expression to that of cultures in exponential phase in YPD medium with the standard glucose concentration with or without LiCl present (new **Fig. S9**). In high glucose, we observed higher expression of ribosomal proteins and lower expression of the respiratory chain components compared to the LiCl control (**Fig. S9A**); there is no indication of a nitrogen starvation response in either of the comparisons. Hence, we have removed the previous reference to C:N ratio from the corresponding paragraph in lines 260-271 and now mention the key results from Fig. S9 in this paragraph.

Can the transcript be engineered so that 100% of the transcripts retain the

intron perhaps by altering the splice site sequence? This would allow testing of whether the bimodal response is due to the intron itself or if the bimodal expression is a result of having both intron and intronless Transcripts.

In our understanding, such a splicing mutant might be attainable, but this is by no means straightforward. The consensus 5' and 3' splice sites and branch sites in the yeast introns would be promising targets for site-directed mutagenesis (Bon et al., 2003, PMID: 12582231). However, their contribution to splicing specificity and efficiency is still a matter of intensive study (Schirman et al., 2021, PMID:34570750). In addition, inefficient splicing might affect the host's transcription rate, RNA stability, or nuclear export (all discussed in Schirman et al. 2021), therefore likely complicating downstream analyses.

Hence, we used an alternative approach based on fluorescence-activated cell sorting (FACS) to address the interesting question if the bimodal expression results from two subpopulations of cells, in one of which the intron is retained while the other one contains only intron-less *RPS22B* transcripts. Specifically, we have sorted the cells by their Rps22B-GFP signal into two groups (with high/low GFP signal) and performed RNA-seq. We found that both Rps22B-high and Rps22B-low cells contain a mixture of intron-retaining and intronless *RPS22B* transcripts (**Fig. S3G**). Rps22B-high cells have slightly lower intron retention and higher CDS transcript level; however, these differences alone appear insufficient to account for the pronounced bimodal protein expression. These results are consistent with a scenario in which the bimodal protein expression is generated by a post-transcriptional mechanism that amplifies the existing transcriptional heterogeneity. We now mention this result in lines 147-156.

Reviewer #2:

Multiple theories exist for why budding yeasts like *Saccharomyces cerevisiae* have retained paralogous ribosomal protein pairs for millions of years after an ancestral whole-genome duplication. Several pairs include one paralog without introns and another with introns that are conserved across species, hinting that differential regulation could provide cellular diversity and differential ribosome functions. In this work, the authors investigate stress-dependent intron retention in the *RPS22B* transcript and its effects on protein expression in yeast. They show that high doses of LiCl stress reduce splicing of the transcript to modulate protein abundance. Interestingly, intermediate doses of LiCl cause cell-to-cell variability in Rps22b protein abundance, and this heterogeneity requires the 5' intron of the transcript. The authors go on to show that intermediate doses of LiCl produce culture heterogeneity, such that some cells in the culture are better able to survive long-term starvation and others instead thrive during regrowth after nutrient repletion. This type of bet hedging is an important strategy in microbes that have evolved to compete in natural settings with unpredictable

environmental shifts.

The manuscript is well presented (although a little over-sold in places in my opinion) and clear, and this is an interesting topic. However, while the authors provide a nice example of how splicing can regulate protein abundance during stress (something that is already known and should be made clearer here), they do not show that heterogeneity in intron retention is causing the phenotypic variation. It is already known that yeast cells vary in their metabolism in a way produces cell-to-cell heterogeneity in how cells respond to starvation (e.g. recent paper by Bagamery et al. in *Current Biology*). Most likely, the heterogeneity arises from some cellular signal above *RPS22B* splicing and that variable signal causes the starvation phenotypes - in other words, *RPS22B* intron retention is likely an output of this heterogeneity rather than the cause. I would be otherwise convinced if the authors showed that the mutant lacking the 5' intron, which lacks the *Rps22b* protein heterogeneity, lost the heterogeneity in starvation responses.

Without a causal link between intron retention and differential starvation response, I think this is nice paper but perhaps not enough sufficiently novel insights to warrant publication in *Nature*.

Additional comments below.

We thank the reviewer for highlighting this issue. We agree that the data in the original manuscript did not fully exclude the possibility that the intron-mediated *Rps22B* protein heterogeneity is merely a marker, rather than the mediator of the differential starvation response. We also agree that the phenotypic effect of the intron deletion is key to making this distinction. However, cell death is generally a stochastic process, often well-approximated by a Poisson process, in which cells die at a given rate, but the survival times of individual cells are exponentially (or otherwise) distributed. This is ultimately due to fundamental stochasticity at the molecular level that cannot be eliminated. Hence, even a mutation that unifies the death rate in a diversified population cannot be expected to eliminate all heterogeneity in survival. We agree though that the phenotypic heterogeneity should decrease in the *RPS22B* 5' UTR intron-deletion mutant if this heterogeneity is mediated by the intron.

As suggested by the reviewer, we have performed new experiments with the 5' UTR intron-deletion mutant to test if the intron deletion affects phenotypic heterogeneity along with protein heterogeneity (new **Fig. 3K, S7 and S8**). Time-lapse microscopy assays revealed that deleting the 5' UTR intron in *RPS22B* increases the death rate under starvation (**Fig. 3K, Fig. S7**) and largely eliminates the phenotypic heterogeneity coupled to *Rps22B* protein heterogeneity (**Fig. S7E**). Please note that since only a minority of the population (~15% in wild type) dies within the time frame of the single-cell experiment, any intermediate increase in mean death rate, even if caused by a loss of heterogeneity in death rates, would superficially look like an increase in survival heterogeneity (**Fig. S7B and D**). The heterogeneity in survival times, which we

observed by CFU counting in a longer population-level experiment, is clearly reduced in the intron-deletion mutant compared with wild type (**Fig. S8A**). These observations are consistent with the disappearance of the slower-dying subpopulation, as expected due to the loss of the Rps22B-low subpopulation in the 5' UTR intron-deletion mutant (**Fig. 2B**). Thus, these results confirm that the *RPS22B* 5' UTR intron mediates not only Rps22B protein heterogeneity, but also the phenotypic heterogeneity. We now explain these new results in lines 229-237.

1) The growth regime should be made clearer. Personally, I don't see the rationale of this method and really the main result is that LiCl alone affects intron retention. At any rate, a clearer description that the "gradient" is not physical but rather a series of cultures with different doses would be useful.

The rationale for isogrowth profiling is to experimentally control for non-specific gene expression changes due to stress and growth inhibition, so that the observed changes can be attributed to the specific perturbation. Without the use of isogrowth profiling in this work, it would not have been possible to determine if the changes in the intron retention rate of ribosomal protein genes are due to the effect of LiCl or due to general growth inhibition (the LiCl treatment resulted in 50% inhibition of growth). Notably, reviewer #3 requested additional controls using isogrowth profiling (see comment 1 of reviewer #3). Also note that the Rps22B bimodal expression pattern could have eluded us had we used only LiCl and cycloheximide at their IC_{50} s, and none of the intermediate conditions in the antiparallel gradient setup (**Fig. 2A**). To avoid any confusion about this approach, we have revised the explanation of the rationale for using the growth regime in the Introduction and Results section (lines 61-68, 81-84). As suggested by the reviewer, we further specified that the two-drug concentration gradient is not a physical gradient in space but a discretized gradient in a series of well-mixed cultures without spatial structure (lines 65-66, 83).

2) In Fig 2D, what does "extent" of bimodality refer to? The median Rps22b expression on the x-axis is described as "for ... wells" but it seems to me that this should be the median abundance per cell? High heterogeneity can give a intermediate culture-level abundance if half the cells have high protein production and half have low.

We thank the reviewer for pointing out this issue. The caption of Fig. 2 previously contained an incorrect reference for the extent of bimodality, which was defined in Fig. S4B of the original manuscript. To avoid any confusion about this definition, we have now included it in **Fig. 2D**. The previous formulation in the figure caption regarding the median was unfortunate and indeed was meant to convey the meaning of median expression of protein across all cells in a given well. We corrected this by changing the word order in the caption of Fig. 2.

3) The authors do a nice job showing the heterogeneity in protein levels. I was left wondering if there is heterogeneity in intron retention in the same cultures (as opposed to low levels of splicing in all cells and

heterogeneity comes from mass-action differences in translation initiation). This could be addressed with single-molecule FISH against the intron in individual cells.

This question is interesting to further elucidate the mechanism underlying the observed heterogeneity in protein levels. We addressed this point using an alternative approach based on cell sorting (FACS), which enables us to detect any heterogeneity in intron-retention in the two subpopulations (Rps22B high/low) and, at the same time, further characterize these two subpopulations phenotypically. Specifically, we have sorted the Rps22B-GFP high and Rps22B-GFP low cells and performed RNA-seq for both subpopulations. We found that both Rps22B-high and Rps22B-low cells contain a mixture of intron-retaining and intronless *RPS22B* transcripts (new **Fig. S3G**). Rps22B-high cells have higher CDS transcript level and slightly lower intron retention rate; however, these differences alone are insufficient to account for the strongly bimodal protein expression. This observation is consistent with a scenario in which the bimodal protein expression is generated by a post-transcriptional mechanism that amplifies the existing transcriptional heterogeneity. We have revised the manuscript to clarify that heterogeneity in intron retention and *RPS22B* transcript level likely contribute to the bimodal expression of the protein but do not fully explain it (lines 147-156).

4) Please list LiCl concentrations in mM so as to more easily compare to other papers that cite concentration.

The LiCl concentrations in Tables 1 and 2 are now reported as both mass and molar concentrations to facilitate comparison with the published literature.

Reviewer #3:

Intron-mediated induction of phenotypic heterogeneity by Lukacisin et al. describes a new phenomenon where stress causes the retention of an intron in a ribosomal protein transcript and thus presumably affects ribosome level. They go on to show that this bimodality can lead to a bet-hedging strategy that has a fitness trade-off depending on environmental conditions. Overall, this is an interesting study with important implications. To me, the critical question is whether this phenomenon is real. While I am largely convinced, I have a couple reservation and experiments I would like to see that would help convince me that the regulation isn't a complex artifact.

1). Is this intron retention or more nascent transcription? In theory, S1C was meant to address this but it compared LiCl to no drug treatment which convolutes growth rate. I think the isogrowth profiling is critical and need to be maintained here. S1C should be redone but compared to cycloheximide at

the same growth condition. Relatedly, give the importance of really solidly showing the phenomenon is real, I think the paper would be greatly strengthened by quantitation of the intron/exon ratio in the polyA fragment. This would more directly detect if this was intron retention or just higher transcriptional rates or less stable mature transcripts. This could be done by sequencing or qPCR.

We agree with the reviewer that the observed increase in intron retention under LiCl is central to our results and needs to be carefully validated. Following the reviewer's suggestion, we have redone Fig. S1C for cycloheximide (new **Fig. S1E**). Unfortunately, growth inhibition by cycloheximide elicits an increase in transcription of ribosomal genes even compared to uninhibited growth; hence, this comparison is not fully conclusive. Therefore, as suggested, we have also performed polyA sequencing of a strain inhibited by LiCl at IC₅₀ and observed an increase in intronic reads similar to that observed for the total RNA fraction (new **Fig. S2B**). These data corroborate that what we observe is intron retention rather than increased nascent transcription. As additional controls for the specificity of this effect, we also analyzed changes in intron retention using polyA-RNA-sequencing for cultures inhibited by two unrelated drugs: the sphingolipid synthesis inhibitor myriocin and the ergosterol synthesis inhibitor fenpropimorph. Neither of these drugs causes an increase in intron retention (**Fig. S2B**). We now mention these additional experiments in lines 95-98.

2) Figure 1C and 1D were a bit tough to understand at first. It wasn't clear at first that all intron were plotted. Given the density I wouldn't be able to tell if the '+' 'x' symbols were non-uniformly distributed. The x-axis was a bit confusing and should probably be labeled as Intron density/exon density. I actually found S1A super helpful and almost feel that part of that data should be incorporated in the main figure.

We thank the reviewer for these constructive suggestions to improve the presentation of our work. To make these results more easily accessible, we have added the cycloheximide condition from Fig. S1A to Fig. 1C as suggested. We have also adjusted the symbols in Fig. 1C to make any potential non-uniform distributions easier to perceive. Further, we have amended the axes labels and now mention explicitly in the caption that all nuclear introns are plotted.

I really wanted to see a direct plot of intron retention versus transcript level. Looking at the plots in S1A made me worry that this could just be more nascent transcripts.

We performed this analysis as suggested by the reviewer and found no indication that intron retention is caused by an increase in absolute transcript level (new **Fig. S1D**). In contrast, intron retention is anti-correlated with transcript level, most likely reflecting increased degradation of intron-retaining transcripts (Danin-Kreiselman, 2003, PMID:12769851). We now mention this observation in lines 94-95 and 136-142.

3) The mutant data around intron deletions are strong. Some direct quantification of transcript level in Fig 2 would probably help make this

stronger along.

We have now quantified the transcript level in the 5' UTR intron deletion mutant. The absence of the 5' UTR intron clearly prevents the cell from downregulating the *RPS22B* transcript under LiCl stress. These data are now included in the new **Fig. S3G**.

4) Where do the long and short starvation times come from? The difference doesn't seem that large. What is the time frame of intron retention during starvation? Shouldn't that be what sets the short and long times. A time course of the change in intron retention would be nice and then you could use that to let the cells tell you what is the relevant time scale (unless there is some good exterior justification for the chosen starvation times)

We selected the starvation times empirically while working with yeast in the microfluidic system. These times were chosen such that an intermediate number of cells died in the long starvation condition or budded after the short starvation. The goal of this approach was to make quantitative changes in the phenotypes between cells with high and low Rps22B expression detectable. We now explain this more clearly in lines 507-509.

We agree with the reviewer that these results should not depend on the exact choice of these time scales, nor on other idiosyncrasies of this assay. The intron retention time frame is indeed an interesting aspect in its own right. However, for the functional analyses, our new observation that intron retention does not correlate strongly with protein-expression bimodality (new **Fig. S3G**) suggested that the time scale of intron retention is unlikely to set the relevant time scale for these phenotypes. Instead, we performed a new assay focused on the Rps22B protein expression levels to test phenotypic differences between Rps22B-low and Rps22B-high subpopulations. Specifically, we separated these two subpopulations using FACS and measured time courses of survival under constant starvation using CFU counts (new **Fig. 3G-H**). For short starvation time, survival of the Rps22B-low and Rps22B-high subpopulations is comparable, while with increasing starvation time the Rps22B-high population dies significantly faster than the Rps22B-low population (new **Fig. 3H**). Moreover, measuring growth curves in fresh medium directly after sorting the cells revealed that the Rps22B-high population grows significantly faster after a short starvation phase (new **Fig. 3I,J**), which occurred during the sorting procedure. These results corroborate our previous observations from the single-cell microscopy assay using an independent approach. Notably, the new survival assay (**Fig. 3H**) no longer depends on any arbitrary choice of starvation time. We now explain these additional experiments and results in lines 224-226.

5) The difference in budding in 3F doesn't look that overwhelming give the variance. Did you check other phenomenon? Was this multiple hypothesis corrected? I worry that this might not be significant. Maybe plotting 3F as a beeswarm or similar plot so we can see all the data points would help some.

We thank the reviewer for this helpful comment. We did not find the beeswarm plot effective to visualize the data in Fig. 3F given the rather unequal sizes of the two groups. Instead, we

replotted these data as cumulative distributions (new **Fig. 3F**), allowing easy assessment of the distribution of individual data points along the entire distribution. To determine statistical significance, we used a Wilcoxon rank-sum test (a.k.a. Mann–Whitney *U* test) with the null hypothesis that the Rps22B expression distribution is equal for the two groups (e.g. budded vs. non-budded for the shorter starvation assay). Since we do not formulate alternative null hypotheses for this dataset, we did not perform multiple hypothesis correction. However, we agree that this result is sufficiently critical to warrant further validation. To further strengthen this result, we have performed new experiments using sorted populations (*cf.* our reply to the previous point). We found that the Rps22B-GFP-high population has a fitness advantage upon replenishment of the growth medium that manifests itself in a higher growth rate, which is even detectable at the population level (new **Fig. 3I,J**). We have amended the main text accordingly (lines 219-226).

Minor

1) The transition to the bistability was a bit abrupt. I would recommend some statement around line 108 that more clearly states how the 5-intron retention would affect level, e.g. leads to a frame shift that leads to an early stop. This would then make it clear that looking at single cell protein levels would be a natural way to follow the phenomenon in multiple Conditions.

We concur that the transition to investigating single-cell protein levels deserves more explanation. Note that the 5' UTR intron is upstream of the coding region. Thus, its retention cannot introduce frameshifts. This intron further contains a stop codon just before the 3' splice site, so its retention cannot lead to an alternative reading of the protein starting more upstream. Rather, its retention likely preserves a dsRNA structure that recruits proteins responsible for RNA degradation (Danin-Kreiselman et al., 2003, PMID: 12769851). We have now included this rationale for how the 5' UTR intron retention leads to an off state in the Results section describing Fig. 2 (lines 140-142).

2) Should 160-176 come before 142-157? I ended up re-reading it in that order when I went through it a second time.

We agree with the reviewer that the suggested order makes the presentation clearer and have rearranged these parts accordingly.

Reviewer Reports on the First Revision:

Referees' comments:

Referee #1 (Remarks to the Author):

The resubmission by Lukacisin et al contains a significant amount of new experimental work that further supports the authors' claim that a 5' UTR in the yeast ribosome protein gene, Rps22B, confers gene expression and phenotypic heterogeneity under specific stress conditions. This strengthens the argument that this non-coding transcript element underlies a bet-hedging strategy in yeast that the authors show operates in response to specific stress conditions (LiCl and stationary phase in very high glucose). The 5'-UTR appears to mediate differential survival in these conditions. The authors have responded to each of the specific comments I raised regarding the initial submission. Importantly, in a new experiment the authors fused the 5' UTR to a GFP gene and show that it is necessary and sufficient to confer bimodal gene expression in the same condition.

Although the study identifies the phenomenon of intron-mediated heterogeneity, the mechanism by which intron retention confers this heterogeneity remains unknown. The authors sorted high and low expressing Rps22B cells and found that both subpopulations contain intron-retaining and intronless RPS22B transcripts. This suggests that there is not a simple relationship between retention of the intron and protein expression in single cells and some post-transcriptional mechanism must underlie the observed bimodal expression of RPS22B in individual cells. Presumably, determination of this mechanism will be the subject of future research.

Referee #2 (Remarks to the Author):

The authors have added a number of new experiments to clarify and extend the manuscript. The topic of heterogeneity in yeast stress responses is interesting and the connection to differential intron retention is a nice extension, but my main question remains if this is enough new insight for publication in Nature.

This is not meant to be critical of the paper, I think it has several nice results, some known before but here connected: that intron retention / splicing controls protein abundance during stress, that population heterogeneity includes heterogeneity in spliced RP transcript and encoded protein abundance, and that this heterogeneity correlates with heterogeneity in growth rate and survival after starvation (Figure 3). I still believe that this heterogeneity is output of the cellular signaling system rather than the sole cause of the starvation survival (see below), and since links between metabolic heterogeneity and bet hedging are already known, the question is what are the most novel aspects of the paper, besides being a nice new example of heterogeneity in a stress response.

The authors added Figure S7 to show that the lack of 5'UTR and intron in RPS22B causes heterogeneity in stress survival. I believe that inappropriate over-production of ribosomal proteins is likely detrimental during stress, so loss of the UTR and the resulting higher protein abundance likely

does limit fitness. The question is if this is really causing or contributing to bimodality in starvation survival aside of just being deleterious. I had a few remaining questions:

1) I was not clear what the rationale was for mCherry being a “cell integrity” marker. Is it meant to reflect on cell lysis? Loss of proteostasis? Loss of ability to make new protein?

2) The manuscript states that the 5’UTR-intron mutant “largely” ablates heterogeneity in protein abundance; while Figure S7B right panel shows a skew toward higher pre-starvation Rps22B-GFP levels, there are still many cells with low expression at levels seen in wild-type cells. But in Figure S7E, a large fraction of these “low” expressing cells still die. Probably that is because these are dynamic systems and mucking with Rps22B levels is deleterious and hard to score in a single time point, but it seems to go against the conclusions of the authors that low Rps22B is advantageous.

Referee #3 (Remarks to the Author):

As stated in the original review I thought the phenomenon described was interest with important implications but had not yet been convinced that the phenomenon was not a complex artifact. The other reviewers felt similarly.

The authors in this revisions have done an extensive job addressing why comments and the new questions that arose for me when I read the other reviewers comments.

I now believe the authors have established that this phenomenon is real and would support its publication.

Issues:

1. Article file to be re-supplied in docx format
2. Please reduce subheadings to 40 characters (with spaces) or less
3. Please remove all main figures from the article file and re-supply individually in EPS, AI PS, PDF, PPT, CDR or XLS (only for graphs) format
4. Please move the figure titles and legends to the end of the main article text after references
5. Please remove tables from the methods – these can be re-supplied as Supplementary information or Extended data
6. Please include a competing interest statement in the article file
7. Please remove all supplementary information (both figures and text) and re-supply it in a separate word docx as a SI Guide
8. The SI figures should be converted into Extended Data (these will appear online in the same file as the main text whereas SI is a separate file, and should not contain key results)
9. References need to be split into main text and methods, numbers should be continuous – note that the main text references should not exceed 50
10. Please reduce the figure 3 legend to 300 words or less
11. Please ensure text in all figures is at least 5pt when resized to the stats sizes
12. Please reduce the depth of Figure 3 by at least 2cm (it is above 17cm in depth when resized to 12cm in width)

Author Rebuttals to First Revision:

Referee #1 (Remarks to the Author):

The resubmission by Lukacisin et al contains a significant amount of new experimental work that further supports the authors' claim that a 5' UTR in the yeast ribosome protein gene, Rps22B, confers gene expression and phenotypic heterogeneity under specific stress conditions. This strengthens the argument that this non-coding transcript element underlies a bet-hedging strategy in yeast that the authors show operates in response to specific stress conditions (LiCl and stationary phase in very high glucose). The 5'-UTR appears to mediate differential survival in these conditions. The authors have responded to each of the specific comments I raised regarding the initial submission. Importantly, in a new experiment the authors fused the 5' UTR to a GFP gene and show that it is necessary and sufficient to confer bimodal gene expression in the same condition.

Although the study identifies the phenomenon of intron-mediated heterogeneity, the mechanism by which intron retention confers this heterogeneity remains unknown. The authors sorted high and low expressing Rps22B cells and found that both subpopulations contain intron-retaining and intronless RPS22B transcripts. This suggests that there is not a simple relationship between retention of the intron and protein expression in single cells and some post-transcriptional mechanism must underlie the observed bimodal expression of RPS22B in individual cells. Presumably, determination of this mechanism will be the subject of future research.

We thank the reviewer for appreciating the additional experimental validation we performed. We concur that testing the expression of GFP fused to the 5' UTR, which was initiated by the reviewer's suggestion, was a key experiment to constrain the mechanism underlying the observed bimodality. We also agree that the detailed molecular mechanism of how the intron retention leads to the bimodal protein expression remains to be fully elucidated and should be the subject of future studies.

Referee #2 (Remarks to the Author):

The authors have added a number of new experiments to clarify and extend the manuscript. The topic of heterogeneity in yeast stress responses is interesting and the connection to differential intron retention is a nice extension, but my main question remains if this is enough new insight for publication in Nature.

We thank the reviewer for appreciating the extension of heterogeneous yeast stress responses to differential intron retention. This novel link not only adds to the list of molecular mechanisms yeast cells use to create phenotypic heterogeneity, but also opens up new research avenues for a sizeable group of evolutionary biologists currently studying the functional role of duplicated genes. Besides this main finding of our study, we also introduce

single-cell isogrowth profiling, which we believe can facilitate discovery in the field of heterogeneous stress responses. Additionally, we find intriguing quantitative relationships between growth rate and the amount of stress that elicits maximum bimodality – an observation that will be interesting for further investigation of bet-hedging architectures by systems biologists and biophysicists. Last but not least, our finding of the specific heterogeneity pattern of Rps22B in hyperosmotic glucose medium, different to ionic osmotic stress and absent in standard laboratory glucose medium, underscores the importance of using the hyperosmotic glucose condition in phenotypic screens by the broad yeast genetics community. Therefore, we believe that our work will appeal to a broad audience and publication of our study in a highly visible journal will benefit the mentioned scientific communities.

This is not meant to be critical of the paper, I think it has several nice results, some known before but here connected: that intron retention / splicing controls protein abundance during stress, that population heterogeneity includes heterogeneity in spliced RP transcript and encoded protein abundance, and that this heterogeneity correlates with heterogeneity in growth rate and survival after starvation (Figure 3). I still believe that this heterogeneity is output of the cellular signalling system rather than the sole cause of the starvation survival (see below), and since links between metabolic heterogeneity and bet hedging are already known, the question is what are the most novel aspects of the paper, besides being a nice new example of heterogeneity in a stress response.

We concur with the reviewer that it is possible that the heterogeneity in splicing and protein abundance is not the only cause of the population heterogeneity. In fact, we do not claim that this heterogeneity is the sole cause of the observed heterogeneity in survival. We have now amended the corresponding statement in the *Discussion* to make this clearer (lines 251-252). What we do conclude is that the intron contributes to the establishment of the survival heterogeneity. We demonstrate this through experiments with intron deletions, which showed that phenotypic heterogeneity is reduced in the intron deletion mutants.

The authors added Figure S7 to show that the lack of 5'UTR and intron in RPS22B causes heterogeneity in stress survival. I believe that inappropriate over-production of ribosomal proteins is likely detrimental during stress, so loss of the UTR and the resulting higher protein abundance likely does limit fitness. The question is if this is really causing or contributing to bimodality in starvation survival aside of just being deleterious. I had a few remaining questions:

1) I was not clear what the rationale was for mCherry being a “cell integrity” marker. Is it meant to reflect on cell lysis? Loss of proteostasis? Loss of ability to make new protein?

We thank the reviewer for pointing out that the rationale for this was not clear. We observe that cells undergoing lysis lose the cytoplasmic mCherry signal, consistent with loss of membrane integrity and release and dilution of cytoplasmic contents into the surrounding medium. This approach was previously used to successfully follow death by starvation in prokaryotic cells (Lowder et al., 2000, PMID: 10919764). This rationale is now clarified in the *Methods* and in the caption of **Extended Data Fig. 7b**.

2) The manuscript states that the 5'UTR-intron mutant “largely” ablates heterogeneity in protein abundance; while Figure S7B right panel shows a skew toward higher pre-starvation Rps22B-GFP levels, there are still many cells with low expression at levels seen in wild-type cells. But in Figure S7E, a large fraction of these “low” expressing cells still die. Probably that is because these are dynamic systems and mucking with Rps22B levels is deleterious and hard to score in a single time point, but it seems to go against the conclusions of the authors that low Rps22B is advantageous.

We thank the reviewer for highlighting this nuanced finding. Indeed, in the microfluidic experiment, we saw cells that, even in the absence of the *RPS22B* 5'UTR intron, showed low expression levels of the Rps22B protein; however, these were much less frequent than in the WT. This finding supports that the 5'UTR intron is important for Rps22Bp downregulation, while suggesting that additional mechanisms of Rps22Bp regulation might be at play. We agree though that one could have hypothesised that the Rps22Bp-low cells from the intron deletion strain would match the higher survival phenotype observed in the Rps22Bp-low cells from the WT strain, since they, too, were able to downregulate the Rps22B protein. However, our results (Fig. S7E, now **Extended Data Fig. 7e**) show that this hypothesis is wrong, and the intronless Rps22Bp-low cells survive the starvation relatively poorly. One can speculate that the alternative means of Rps22B downregulation are not beneficial in this setting due to different temporal dynamics, or that the fitness phenotype is dependent on both protein level and the retention of the intron itself; in any case, this observation does not weaken, but rather strengthens our central finding of phenotypic heterogeneity that is mediated by an intron. We have now extended the explanation of the effects of the intron deletion in the main text to more explicitly make this point (lines 209-212).

Referee #3 (Remarks to the Author):

As stated in the original review I thought the phenomenon described was interest with important implications but had not yet been convinced that the phenomenon was not a complex artifact. The other reviewers felt similarly.

The authors in this revisions have done an extensive job addressing why comments and the new questions that arose for me when I read the other reviewers comments. I now believe the authors have established that this phenomenon is real and would support its publication.

We thank the reviewer for their encouraging comments regarding the interest and implications of our finding, as well as for the specific suggestions for experimental and analytical validations made during the previous round of review, which helped us exclude the possibility that our findings are the result of a complex artefact.